# Rethinking Neural Combinatorial Optimization for Vehicle Routing Problems with Different Constraint Tightness Degrees

**Fu Luo**[1,2], **Yaoxin Wu**[3], **Zhi Zheng**[4], **Zhenkun Wang**[1,2]*

[1] School of Automation and Intelligent Manufacturing,
Southern University of Science and Technology, Shenzhen, China
[2] Guangdong Provincial Key Laboratory of Fully Actuated System Control Theory and Technology,
Southern University of Science and Technology, Shenzhen, China
[3] Department of Industrial Engineering and Innovation Sciences,
Eindhoven University of Technology, Eindhoven, The Netherlands
[4] School of Computing, National University of Singapore, Singapore
`luof2023@mail.sustech.edu.cn, y.wu2@tue.nl,`
`zhi.zheng@u.nus.edu, wangzhenkun90@gmail.com`

## Abstract

Recent neural combinatorial optimization (NCO) methods have shown promising problem-solving ability without requiring domain-specific expertise. Most existing NCO methods use training and testing data with a fixed constraint value and lack research on the effect of constraint tightness on the performance of NCO methods. This paper takes the capacity-constrained vehicle routing problem (CVRP) as an example to empirically analyze the NCO performance under different tightness degrees of the capacity constraint. Our analysis reveals that existing NCO methods overfit the capacity constraint, and they can only perform satisfactorily on a small range of the constraint values but poorly on other values. To tackle this drawback of existing NCO methods, we develop an efficient training scheme that explicitly considers varying degrees of constraint tightness and propose a multi-expert module to learn a generally adaptable solving strategy. Experimental results show that the proposed method can effectively overcome the overfitting issue, demonstrating superior performance on the CVRP and CVRP with time windows (CVRPTW) with various constraint tightness degrees. The code is available at https://github.com/CIAM-Group/Rethinking_Constraint_Tightness.

## 1 Introduction

The vehicle routing problem (VRP) is an important category of combinatorial optimization problems and is encountered in many real-world applications, including transportation [1], navigation [2], and robotics [3]. Due to the NP-hard nature [4], it is prohibitively expensive to obtain exact VRP solutions. Traditional methods typically employ problem-specific heuristic algorithms, which require extensive domain-specific expertise. As an alternative, neural combinatorial optimization (NCO) methods have received widespread attention recently [5–8]. These methods can automatically learn problem-solving strategies from data using neural networks. Some advanced NCO methods can even obtain near-optimal VRP solutions without the need for expert knowledge [9–17].

VRP instances are typically defined with some numeric constraints. For example, the solution of a capacity-constrained VRP (CVRP) should meet the capacity constraint of each vehicle, and the

---

*Corresponding author

Table 1: Comparison results of three representative NCO models on CVRP100 instances with in-domain capacity value (C=50) and out-of-domain capacity values (C={10,500}). "Gap" quantifies the percentage difference between a method's solution and the ground truth provided by the heuristic solver HGS [21]. "Expansion" measures the ratio of the increase in the gap relative to the in-domain baseline (C=50).

| | In-Domain | Out-Of-Domain | | | |
|---|---|---|---|---|---|
| | C=50 | C=10 | | C=500 | |
| | Gap | Gap | Expansion | Gap | Expansion |
| POMO | 3.66% | 20.26% | 5.54× | 34.12% | 9.32× |
| LEHD | 4.22% | 36.56% | 8.66× | 6.82% | 1.62× |
| BQ | 3.25% | 18.02% | 5.54× | 8.16% | 2.51× |

solution of a CVRP with time windows (CVRPTW) should visit each customer in its admissible time windows. Existing NCO methods usually train their model on problem instances with fixed numeric constraint settings. For example, the AM [18], POMO [9], BQ [19], and LEHD [20] train and test their models on CVRP instances with 100 customers (CVRP100), assuming a fixed vehicle capacity of 50 and an average customer demand of 5. In this case, the capacity C=50 corresponds to a constraint of relatively moderate tightness. However, when dealing with CVRP100 instances with some extreme constraints (e.g., C=10 as a tight constraint and C=500 as a loose constraint), the performance of these NCO models significantly degenerated, as illustrated in Section 3.

Current NCO research has been deluded by the seemingly promising results obtained on instances with certain specific tightness, lacking a comprehensive study of how constraint tightness affects model performance. To fill this gap, this paper takes CVRP as an example and performs an empirical analysis of the NCO performance under different capacity constraint tightness. We evaluate three recently proposed NCO methods (i.e., POMO, LEHD, and BQ) that are trained using CVRP instances with fixed customer demands but different capacity values, and the results are presented in Table 1. As can be seen from the table, when the CVRP has extremely tight or loose capacity constraints (C=10 and 500), the optimality gaps of the NCO methods deteriorate by at least 5.5× and 1.6×, demonstrating severe performance degradation. We believe that these NCO models overfit the training instances with a specific capacity tightness, i.e., C=50, and thus exhibit limited generalization ability across instances with other capacity values. In this paper, we reveal the reasons behind the performance degradation by leveraging the problem similarity between CVRP and two other VRP variants, i.e., the traveling salesman problem (TSP) and the open vehicle routing problem (OVRP). More details are provided in Section 4.1.

Beyond that, this paper presents a simple yet effective training scheme to enhance the performance of existing NCO models on instances of varying constraint tightness degrees. The training scheme enables models to learn a policy that is effective across a wide range of changing constraint tightness. To further enhance adaptability, we propose a multi-expert module to augment neural architectures of NCO models, in which each expert specializes in distinct ranges of constraint tightness degrees. Experiments on CVRP and CVRPTW validate the effectiveness of our method across instances with varying tightness degrees.

Our contribution can be summarized as follows:

- We revisit the training setting of existing NCO models and find that models trained under these settings may overfit certain specific constraint tightness. They suffer from severe performance degradation when applied to VRP instances with out-of-domain tightness degrees.

- We propose a simple yet efficient training scheme that enables the NCO model to be efficient on a broad spectrum of constraint tightness degrees. Moreover, we propose a multi-expert module to enable the NCO model to learn a more effective policy for coping with diverse constraint tightness degrees.

- Through extensive experiments on CVRP and CVRPTW, we validate the effectiveness and robustness of our method in solving instances with constraint tightness ranging from tight to extremely loose. The ablation results demonstrate that the proposed training scheme and multi-expert module are efficient in enhancing existing NCO models.

Table 2: Performance of recent advanced NCO models on CVRP100 instances with different capacities. The lowest optimality gap of each model across all datasets is in bold.

| | CVRP100 | | | | | | | |
| | C=10 Gap | C=50 Gap | C=100 Gap | C=200 Gap | C=300 Gap | C=400 Gap | C=500 Gap | Varying Capacities Avg. Gap |
|---|---|---|---|---|---|---|---|---|
| HGS [21] | 0.00% | 0.00% | 0.00% | 0.00% | 0.00% | 0.00% | 0.00% | 0.00% |
| AM [18] | 45.16% | **7.66%** | 12.85% | 21.63% | 26.26% | 25.35% | 27.23% | 23.76% |
| POMO [9] | 20.26% | **3.66%** | 11.17% | 22.45% | 29.56% | 30.80% | 34.12% | 21.72% |
| MDAM [22] | 7.47% | **5.38%** | 12.47% | 21.13% | 24.31% | 22.55% | 23.91% | 16.75% |
| BQ [19] | 18.02% | **3.25%** | 3.48% | 4.53% | 6.53% | 6.54% | 8.16% | 7.22% |
| LEHD [20] | 36.56% | 4.22% | 4.87% | **4.18%** | 5.78% | 4.92% | 6.82% | 9.62% |
| ELG [23] | 10.44% | **5.24%** | 10.91% | 18.70% | 22.81% | 21.84% | 24.05% | 16.29% |
| INViT [24] | 17.44% | **7.85%** | 11.12% | 14.36% | 15.86% | 12.61% | 13.71% | 13.28% |
| POMO-MTL [25] | 13.62% | **4.50%** | 8.20% | 10.40% | 14.12% | 13.44% | 15.57% | 11.41% |
| MVMoE [26] | 10.72% | **5.06%** | 10.52% | 17.63% | 21.80% | 21.86% | 16.00% | 17.24% |

## 2 Preliminaries

### 2.1 Problem Definition

A VRP instance $S$ can be defined on a graph $\mathcal{G} = (\mathcal{V}, \mathcal{E})$, where $\mathcal{V}$ denotes the node set and $\mathcal{E} = \{(v_i, v_j) | v_i, v_j \in \mathcal{V}, v_i \neq v_j\}$ denotes the edge set. The nodes from $v_1$ to $v_n$ represent $n$ customers, and some VRPs (e.g., CVRP, OVRP) contain a depot $v_0$ with $\mathcal{V} = \{v_i\}_{i=0}^n$. In this work, we assume an unlimited fleet of homogeneous vehicles starting from and returning to the depot to serve all customers. This follows the classical VRP formulation used in seminal NCO papers [18, 9].

The solution of a VRP instance is a tour $\boldsymbol{\pi}$, which is a permutation of the nodes. Given an objective function $f(\cdot)$ of solutions, solving the VRP instance is to search for the optimal solution $\boldsymbol{\pi}^*$ with minimal objective function value (e.g., Euclidean distance of the tour) under certain problem-specific constraints (e.g., visiting each node exactly once in solving the traveling salesman problem (TSP)). Therefore, a VRP can be formulated as follows:

$$\boldsymbol{\pi}^* = \underset{\boldsymbol{\pi} \in \Omega}{\text{minimize}} \quad f(\boldsymbol{\pi}) \tag{1}$$

where $\Omega$ is the set of all feasible solutions, with each solution $\boldsymbol{\pi}$ satisfying the problem-specific constraints. The commonly studied VRPs are elaborated below.

**TSP** The TSP is defined with $n$ customer nodes. It minimizes the length of a tour starting from an arbitrary customer, visiting each customer exactly once, and returning to the starting customer.

**CVRP** The CVRP contains a depot node $v_0$ and $n$ customer nodes. Each customer node $i$ has a demand $\delta_i$ to fulfill, and a feasible tour in CVRP is constituted by a set of sub-tours that start and end at the depot, ensuring the total demand in each sub-tour does not exceed the vehicle capacity. The objective is to minimize the total distance of the sub-tours while satisfying the capacity constraint.

**OVRP** OVRP extends CVRP by removing the requirement for vehicles to return to the depot after servicing customers in each sub-tour. Instead, the vehicles can end their tours at the last served customers, offering more flexibility in practical vehicle routing.

### 2.2 VRP Solution Construction

NCO methods primarily employ encoder-decoder neural networks to learn policies for constructing VRP solutions [18, 9]. Given a VRP instance, the encoder calculates embedding $\mathbf{h}_i$ for each node $v_i$. These embeddings $\{\mathbf{h}_i\}_{i=0}^n$ are then utilized by the decoder to incrementally construct a VRP solution (i.e., a tour) by adding one node at a time to the existing partial solution. Starting from an empty solution, the decoder iteratively selects one unvisited node at each step, appends it to the incomplete solution, and marks it as visited. For example, at step $t$, the incomplete solution is represented as $(\pi_1, \pi_2, \ldots, \pi_{t-1})$, where $\pi_1$ denotes the first node added to the solution and $\pi_{t-1}$ represents the most recently added node. This node-by-node selection process continues until all

nodes are visited, resulting in a complete solution. The encoder-decoder neural networks are often trained by reinforcement learning [18, 9, 23] or supervised learning [27, 28, 19].

## 2.3 Definition of Constraint Tightness

For CVRP, the "constraint tightness" is directly represented by the vehicle capacity value, $C$. A tight constraint corresponds to a low capacity value (e.g., $C = 10$ in our paper). In this scenario, each vehicle can only serve a very limited number of customers, forcing the solution to be composed of many short sub-tours. A loose constraint corresponds to a high capacity value (e.g., $C = 500$). Here, the capacity is so large that it is barely a limiting factor, allowing a single vehicle to serve many or all customers. This makes the problem characteristics approach those of the Traveling Salesman Problem (TSP). Therefore, in the context of CVRP, constraint tightness is inversely proportional to the vehicle capacity $C$.

We also provide a more general definition of constraint tightness independent of the problem features, such as capacity, with details presented in Appendix J

# 3 NCO Performance on CVRP with Different Constraint Tightnesses

To investigate the impact of different constraint tightnesses on model performance, in this section, we conduct an empirical study of advanced NCO models on CVRP instances with different constraint tightnesses.

## 3.1 Experiment Details

**Dataset**    Following the data generator in prior works [18, 9], we generate seven test datasets, each comprising 10,000 CVRP100 instances. Customer demands are drawn uniformly from $\{1, \ldots, 9\}$, so the average customer demand is 5. Vehicle capacities for these datasets are set to 10, 50, 100, 200, 300, 400, and 500, respectively. C=500 represents a loose constraint that is expected to allow a single vehicle to feasibly serve all customers in a single route, while C=10 represents the tight constraint that allows a vehicle to serve only two customers on average. All node coordinates are uniformly sampled within a unit square, so these datasets differ solely in capacity values.

**NCO Methods**    We comprehensively collect different NCO methods for solving CVRP, including POMO [9], BQ [19], LEHD [20], INViT [24], ELG [23], MVMoE [26], AM [18], MDAM [22], POMO-MTL [25]. We adopt their provided pre-trained models, trained on CVRP100 with C=50, for the following evaluation.

**Metrics & Inference**    We evaluate the optimality gap (Gap) across all methods and datasets, which quantifies the percentage difference between a method's solution and the ground truth (generated by the HGS solver [21]). All evaluated NCO methods use the greedy search for inference to present their basic problem-solving ability.

## 3.2 Results and Observations

The experimental results are shown in Table 2. We observe that all NCO methods achieve low optimality gaps on test instances with the same constraint tightness (e.g., C=50) as the training instances. However, their performance drops significantly on instances with unseen loose or tight constraints. For example, the optimality gap of BQ model on the dataset with C=50 is 3.25%, but the gap on the dataset with C=10 drastically expands to 18.02%. POMO also shows a similar trend, with the gap on the dataset with C=50 being 3.66%, and expanding to 34.12% on the dataset with C=500. Overall, compared to the test dataset with the same capacity value as the training set, all evaluated NCO models exhibit at least a 1.7× increase in the average optimality gap (i.e., INViT has an average gap of 13.28%, which is 1.7× larger than 7.85% on the dataset with C=50) when tested on the datasets with different capacities. These results indicate that current NCO models generally overfit to the trained constraint tightness, which causes the learned strategies to be poorly applied to instances with other constraint tightnesses, ultimately significantly degrading their performance.

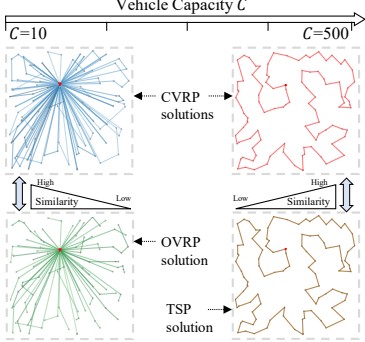

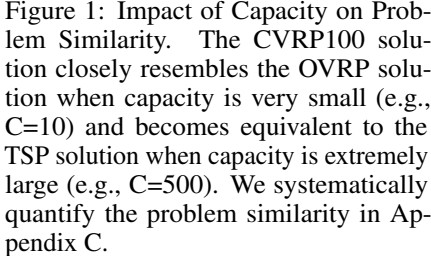

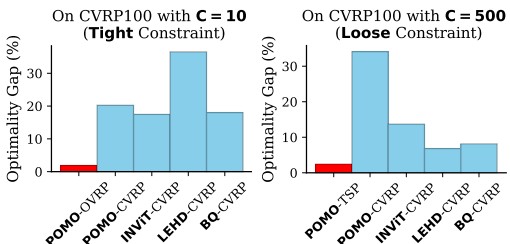

Figure 1: Impact of Capacity on Problem Similarity. The CVRP100 solution closely resembles the OVRP solution when capacity is very small (e.g., C=10) and becomes equivalent to the TSP solution when capacity is extremely large (e.g., C=500). We systematically quantify the problem similarity in Appendix C.

Figure 2: Performance of different NCO methods on CVRP100 instances with extreme constraint tightness degrees (i.e., C=10/500). "-OVRP", "-TSP", and "-CVRP" indicate that the methods are specifically designed to solve OVRP, TSP, and CVRP instances, respectively. A smaller optimality gap indicates better performance.

## 4 Methodology

In this section, we analyze the limitations of current NCO models from the perspective of constraint tightness. To address the performance degradation led by the constraint tightness, we propose two effective modifications to NCO models in terms of the training protocol and neural architecture. These modifications aim to enable the model to learn strategies that can adapt to varying degrees of constraint tightness, thereby improving the performance of NCO models in solving more general VRPs.

### 4.1 Limitations of NCO Methods

As observed in Section 3, NCO models generally fail to retain good performance under varying constraint tightnesses.

When the constraint tightness changes drastically, the optimal solution structure of CVRP undergoes significant transformations, indicating that the problem requires fundamentally different solving strategies. As shown in Fig. 1, under the tight capacity constraint C=10, where the capacity is capped at twice the average customer demand, each sub-tour in the CVRP solution can only serve a limited number of customers (e.g., two or three). In this case, CVRP solutions exhibit structural similarities to OVRP solutions under identical capacity constraints, as sub-tours in their solutions only visit very few customers. That is, when vehicle capacity is severely limited, the problem-solving strategy for CVRP becomes similar to that of OVRP (except that returning to the depot is neglected). Conversely, under loose capacity constraints (e.g., C=500, approaching total customer demand), the vehicles can serve all customers in a few or even a single sub-tour. The solution structure of CVRP resembles that of TSP in this situation. Therefore, TSP-oriented solving strategies could work well for solving CVRP instances with large capacities.

We conduct experiments to validate our conjecture. Specifically, we employ the POMO model specialized for solving OVRP100 instances with C=10 (denoted as POMO-OVRP) to solve CVRP100 instances (with C=10). Also, we employ the POMO model specialized for TSP (denoted as POMO-TSP) to solve CVRP100 instances (with C=500). For comparison, we test several representative NCO methods, including POMO, INViT, LEHD, and BQ, on CVRP100 instances with C=10 and C=500. These models are trained on CVRP100 instances with C=50. The results are shown in Figure 2. We observe that under small capacity, POMO-OVRP outperforms all CVRP models. For example, POMO-OVRP achieves an optimality gap of 1.9%, which is much lower than INViT's

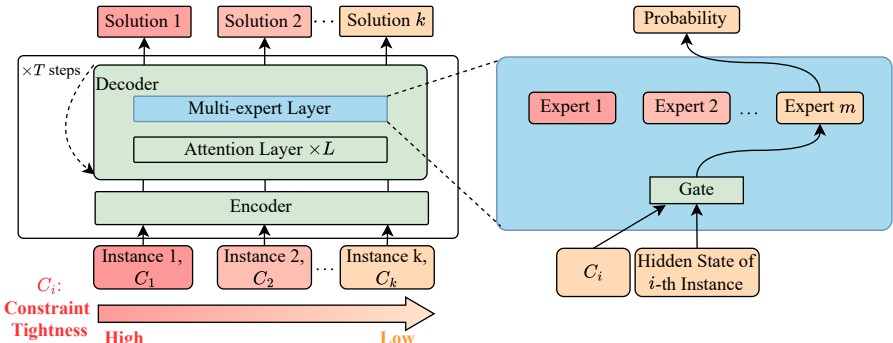

Figure 3: Model structure featuring a multi-expert module. The model adopts a light encoder and heavy decoder architecture, incorporating an effective multi-expert module within the decoder. The multi-expert module consists of a gate mechanism and $m$ expert layers. The gate mechanism selects instances within specific ranges of constraint tightness to their corresponding expert layer for processing. Each expert layer processes the node embeddings from the $L$ stacked attention layers and generates probability distributions for node selection during solution construction.

17.44%. Conversely, for CVRP instances with C=500, the POMO-TSP model with a gap of 2.5% demonstrates superiority over specialized NCO models for CVRP (e.g., LEHD with a gap of 6.8%).

The results corroborate our conjecture that solving strategies for CVRP under different capacities (C={10,50,500}) differ significantly. The NCO models trained on CVRP instances with C=50 cannot be effectively applied to instances with different capacity values, thereby exhibiting significant performance degradation.

## 4.2 Training with Varying Constraint Tightness

In order to address the overfitting issue of existing NCO methods to enable the model to learn effective problem-solving strategies across different degrees of constraint tightness, we propose a straightforward yet effective training scheme, which explicitly exposes the model to the full range of constraint tightnesses. Specifically, let $C_{min}$ and $C_{max}$ denote the minimum and maximum constraint tightness degrees in the target domain (e.g., vehicle capacities in VRPs). During training, the constraint tightness degrees are uniformly sampled from $[C_{min}, C_{max}]$ for each training instance. By continuously changing the constraint tightness of the training data, the NCO model is able to learn to adjust its decision-making strategy in response to varying problem domains, improving its robustness in solving general VRP instances.

In addition, we attempt to assign all instances in a training batch the identical tightness degree, randomly sampled from $[C_{min}, C_{max}]$. This batch-level tightness assignment, compared to the above instance-level assignment, leads to inferior model performance. A more detailed discussion with experiments can be found in Appendix E.

## 4.3 Multi-Expert Module

When training models to handle instances with varying constraint tightness degrees, inconsistencies in problem characteristics may create conflicting optimization directions, leading to suboptimal learning outcomes. To address this challenge, we propose integrating a multi-expert module into existing NCO models, fostering a more general and effective routing strategy across a broader range of constraint tightnesses.

Given the outstanding performance of the Lightweight Encoder-Heavy Decoder (LEHD) model [19, 20], we adopt it as the backbone model to exemplify the implementation of the multi-expert module. In general, LEHD uses $L$ stacked attention layers in the decoder to construct a VRP solution step by step. As shown in Fig. 3, we integrate the multi-expert module after the stacked attention layers in the decoder of LEHD. The multi-expert module consists of an instance-level gate mechanism $G(\cdot)$ and $m$ expert layers $\{E_1, E_2, \cdots, E_m\}$.

Table 3: Comparison results between our model and the existing best model in each dataset. Gaps on C=10, C=50/100, and C=200 to 500 come from MDAM, BQ, and LEHD, respectively.

| | CVRP100 | | | | | | | |
| | C=10 Gap | C=50 Gap | C=100 Gap | C=200 Gap | C=300 Gap | C=400 Gap | C=500 Gap | Varying Capacities Avg. Gap |
|---|---|---|---|---|---|---|---|---|
| Existing Best | 7.47% | **3.25%** | **3.48%** | 4.18% | 5.78% | 4.92% | 6.82% | 5.13% |
| Ours | | **1.54%** | 3.82% | 3.67% | **1.49%** | **0.87%** | **0.75%** | **0.87%** | **1.86%** |

Given an instance with constraint tightness $C_k$, we denote its node embedding matrix after the $L$ stacked attention layers as $H_t = \{h_i^{(L)}\}_{i \in N_a^t}$, where $N_a^t$ represents the number of unvisited nodes at the $t$-th step during solution construction. With $C_k$ and $H_t$ as inputs, the multi-expert module generates the node selection probability matrix $P_t$, which can be formulated as

$$P_t = \text{MEM}(H_t, C_k) = \sum_{i=1}^{m} G(C_k) E_i(H_t). \tag{2}$$

**Gate Mechanism**  Given an instance, the gate mechanism assigns the node embedding matrix $H_t$ to the appropriate expert layer based on the constraint tightness $C_k$. Specifically, the $i$-th expert layer ($i \in \{1, 2, \cdots, m\}$) processes instances with constraint tightnesses in the interval $[(i-1)\beta, i\beta]$, where $\beta = \frac{|C_{max} - C_{min}|}{m}$. Therefore, the gate mechanism can be formulated as a one-hot vector

$$G(C_t) = \begin{cases} 1 & \text{if } (i-1)\beta \leq C_t - C_{\min} < i\beta, \\ 0 & \text{otherwise.} \end{cases} \tag{3}$$

**Expert Layer**  Each expert layer transforms the node embedding matrix $H_t$ into the probability matrix for node selection. To enhance the adaptability to varying constraint tightness, each expert layer comprises $m_e$ stacked attention layers (see Appendix B.1 for details) before the probability prediction. The calculation process within the $i$-th expert layer can be formulated as:

$$\begin{aligned} H_t^{(1)} &= \text{AttnLayer}_{\text{i},1}(H_t), \\ &\cdots \\ H_t^{(m_e)} &= \text{AttnLayer}_{\text{i},\text{m}_e}(H_t^{(m_e - 1)}), \\ P_t &= \text{Softmax}(W_i H_t^{(m_e)} + b_i), \end{aligned} \tag{4}$$

where $W_i$ and $b_i$ are learnable parameters. Different expert layers are specialized in processing different constraint tightnesses, corresponding to more effective VRP strategies within one single NCO model.

## 5  Experiment

In this section, we first evaluate the effectiveness of our method across varying degrees of constraint tightness. Next, we conduct ablation studies on the key components of the proposed method. Finally, we validate the versatility of our method by applying it to CVRPTW.

**Dataset&Baseline**  We reuse the seven test datasets from Section 3.1 to evaluate models. We evaluate the NCO methods mentioned in Section 3.1 as baselines. To make our results more significant, we consistently compare the best NCO model for each dataset.

**Model Setting&Training**  We utilize the LEHD model [20, 19] as the backbone to demonstrate the effectiveness of the proposed method. The embedding dimension is set to $d = 192$, the hidden dimension of the feed-forward layer is set to 512, the query dimension $q$ is set to 16, and the head number in multi-head attention is set to 12. The decoder includes $L = 6$ stacked attention layers and a multi-expert module with $m = 3$ expert layers, each containing $m_e = 3$ layers. Following [20], we use supervised learning to train the model. The training dataset comprises one million CVRP100 instances and their corresponding labeled solutions. Each training instance has a random capacity drawn from $\{10, 11, \cdots, 500\}$, and the labeled solutions are generated using HGS [21] solver. The Adam optimizer [29] is utilized for training the models, with an initial learning rate of 1e-4 and a decay rate of 0.9 per epoch. We train the model for 40 epochs, with the batch size set to 512.

**Metrics&Inference** For comparative analysis, we evaluate the metric of optimality gap (Gap) across all methods and datasets, as done in Section 3.1. All models use the greedy search for inference. The inference time that quantifies a method's computational efficiency is shown in Appendix A. All experiments, including training and testing, are executed on a single NVIDIA GeForce RTX 3090 GPU with 24GB of memory.

## 5.1 Comparison Results

The experimental results are presented in Table 3, which demonstrate that our method outperforms state-of-the-art NCO methods overall. Specifically, our method achieves an average gap of 1.86%, significantly lower than existing NCO models' average best gap of 3.71%. While our method exhibits marginally inferior performance compared to the baselines at C=50 and 100, it outperforms all competing methods in other scenarios (C=10, C=200, and above). Notably, for instances with C=300 to C=500, our method maintains an optimality gap below 1%. These results highlight the effectiveness of our method in solving problems across varying degrees of constraint tightness.

Table 4: Effects of Varying Constraint Tightness training (VCT) and Multi-Expert Module (MEM). The Base Model is only trained on instances with C=50 and without MEM.

|  | CVRP100 | | | | | | | |
|---|---|---|---|---|---|---|---|---|
|  | C=10 Gap | C=50 Gap | C=100 Gap | C=200 Gap | C=300 Gap | C=400 Gap | C=500 Gap | Varying Capacities Avg. Gap |
| Base Model | 45.64% | 3.72% | 4.21% | 3.62% | 5.61% | 5.36% | 7.29% | 10.78% |
| Base Model+MEM | 37.03% | **3.28%** | 4.41% | 2.90% | 4.56% | 4.39% | 6.11% | 8.95% |
| Base Model+VCT | 1.88% | 4.51% | 4.69% | 2.25% | 1.28% | 1.03% | 1.25% | 2.41% |
| Base Model+VCT+MEM | **1.54%** | 3.82% | **3.67%** | **1.49%** | **0.87%** | **0.75%** | **0.87%** | **1.86%** |

Table 5: Results on CVRPTW instances with different degrees of time windows constraint tightness.

|  | CVRPTW100 | | | | | | | |
|---|---|---|---|---|---|---|---|---|
|  | $\alpha = 0.2$ Gap | $\alpha = 0.5$ Gap | $\alpha = 1.0$ Gap | $\alpha = 1.5$ Gap | $\alpha = 2.0$ Gap | $\alpha = 2.5$ Gap | $\alpha = 3.0$ Gap | Varying Time Windows Avg. Gap |
| POMO | 14.95% | 13.11% | **11.39%** | 16.40% | 21.62% | 27.34% | 33.56% | 19.58% |
| POMO+VCT | 9.80% | 12.22% | 12.37% | 14.11% | 13.73% | 12.40% | 11.23% | 12.14% |
| POMO+VCT+MEM | **9.13%** | **11.57%** | 11.91% | **13.30%** | **13.24%** | **11.58%** | **10.37%** | **11.58%** |

## 5.2 Ablation Study

We conduct an ablation study to evaluate the contributions of two key components: 1) varying constraint tightness training (VCT) and 2) the multi-expert module (MEM). We compare three training configurations: the model trained on CVRP100 instances with fixed capacity (C=50) 1) without MEM and 2) with MEM; 3) the model trained on instances with capacities $C \sim$ **Unif**$([10, 500])$ and without MEM; 4) the model trained on instances with capacities $C \sim$ **Unif**$([10, 500])$ and with MEM. We maintain a consistent training data budget across all experimental settings. Results of these models on the test sets are presented in Table 4.

From these results, we can observe that while adding only the MEM reduces the average gap from 10.78% to 8.95%, providing a modest benefit, varying constraint tightness training leads to significant improvements in the model's problem-solving ability on 5 out of 7 datasets, reducing the average gap from 10.78% to 2.41%. However, this approach causes a slight performance degradation on the datasets with C=50 and C=100. The reason may be that the different constraint tightness degrees create conflicting optimization directions, leading to suboptimal learning outcomes. When the multi-expert module is incorporated, the model achieves consistent performance gains across all datasets, reducing the average gap by 23% (from 2.41% to 1.86%). These results demonstrate that while varying constraint tightness training enhances cross-constraint-tightness generalization, the integration of the multi-expert module further strengthens this capability, enabling more robust adaptation to problem instances with varying constraint tightness.

# 6 Versatility

To demonstrate that our proposed method is applicable to other routing problems, we validate it on CVRPTW instances with varying degrees of time window constraint tightness. We provide the CVRPTW definition and introduce a coefficient $\alpha$ to quantify the tightness of the time window constraint coefficient in Appendix F.

We apply our proposed method to the classic POMO model to solve CVRPTW instances with varying degrees of time window constraint tightness. The hyperparameter settings of the POMO model follow the description in its original literature [9]. We train the model for 1,000 epochs using reinforcement learning, with each epoch containing 10,000 samples and a batch size of 64. For training under different constraint tightness degrees, the tightness coefficient for the time window constraints is set to $\alpha \sim \text{Uniform}(0.0, 3.0)$, where $\alpha \to 0.0$, $\alpha = 1.0$, and $\alpha = 3.0$ denote the extremely tight, moderate, and loose constraints, respectively. For the multi-expert module incorporated into POMO, the $i$-th expert layer is specialized in solving instances with time window tightness in the range $[(i-1), i]$. Implementation details for the multi-expert module can be found in Appendix G. We train three POMO models separately: 1) the original POMO model trained with the instances of default time window constraint (i.e., $\alpha = 1.0$), 2) the model with varying constraint tightness training, and 3) the model with varying constraint tightness training and the multi-expert module. These models use greedy search for inference and are evaluated on datasets with tightness coefficients ranging from $\alpha = 0.2$ to $\alpha = 3.0$. The experimental results are presented in Table 5.

From the experimental results, we observe that when $alpha$ changes from 1.0 to 0.2/3.0, the optimality gap of the original POMO model increases from 11.39% to 14.95% and 33.56%, respectively. The introduction of varying constraint tightness training improves the model's performance in all cases except for $\alpha = 1.0$. The augmentation of the multi-expert module further enhances the model performance across all degrees of constraint tightness. These results indicate that in CVRPTW, the NCO method also has the overfitting issue and exhibits significant performance degradation when the constraint tightness changes drastically, and the proposed method can effectively tackle the limitation, enhancing the NCO model for VRP with varying domains of time windows.

# 7 Related Work

## 7.1 NCO on VRPs with Fixed Constraint Tightness

NCO methods primarily learn construction heuristics to arrange input nodes into feasible solutions. The pioneering Ptr-Net [27] introduces a Recurrent Neural Network (RNN)-based architecture to solve TSP through supervised learning. In contrast, Bello et al. [30] employs reinforcement learning to eliminate dependency on labeled solutions. Subsequent work by Nazari et al. [31] extends this framework to solve CVRP. A significant advancement comes with the Attention Model [18], which adopts the Transformer architecture [32] to achieve state-of-the-art performance across various VRPs with up to 100 nodes. This inspires numerous AM-based enhancements [9, 33, 22, 34–37] that narrow the performance gap with classical heuristic solvers. However, these methods share a critical limitation: they are typically trained on VRP instances with fixed constraint tightness. For example, models trained on CVRP100 instances typically adopt a fixed capacity of C=50 [9, 38, 19, 20, 24, 26, 25, 23]. This training paradigm may overfit the NCO models on training instances with certain specific constraint tightness, restricting their adaptability to other cases. As a result, these models suffer severe performance degradation when applied to instances with out-of-domain constraint tightness. Our work systematically investigates the underlying causes of this limitation and provides new insights to tackle it.

## 7.2 NCO on VRPs with Varying Constraint Tightness

Recent efforts have attempted to incorporate capacity variations during training [39, 40]. Among them, Manchanda et al. [39] apply meta-learning [41] to train models on CVRP instances with a few certain capacities $C \in \{10, 30, 40\}$, while Zhou et al. [40] focus on the instance-level large-scale generalization considering different capacities. While these approaches demonstrate improved generalization compared to fixed-capacity training, they exhibit two key shortcomings. First, they lack a comprehensive study of how constraint tightness affects model performance and an in-depth analysis of the cause. Second, they fail to account for extreme scenarios (e.g., C=500), which

may lead to model performance deterioration when solving instances with extremely small or large capacities. Our work revisits these challenges through a systematic analysis of constraint tightness impacts, providing methods to tackle different tightnesses of general VRP constraints (i.e., vehicle capacity and time window).

## 8    Conclusion, Limitation, and Future Work

**Conclusion**    In this work, we revisit the training setting of existing NCO models and empirically reveal that models trained under this setting may overfit instances of specific constraint tightness, thereby suffering from severe performance degradation when applied to instances with out-of-domain tightness degrees. To overcome the limitation, we propose a varying constraint tightness training scheme and a multi-expert module. Extensive experiments on CVRP and CVRPTW validate the effectiveness and robustness of our method in solving instances with constraint tightness ranging from extremely tight to extremely loose. In future work, we plan to apply the proposed method to other NCO models to further explore its generality.

**Limitation and Future Work**    While our model demonstrates strong generalization performance across varying degrees of constraint tightness, it is currently trained using a uniform sampling strategy, where the constraint tightness degrees of training instances are uniformly sampled. A promising direction is to develop an adaptive sampling strategy for further performance enhancement across the entire constraint tightness spectrum. Another important future direction involves extending our method to other COPs, such as scheduling, planning, and bin packing.

## Acknowledgments and Disclosure of Funding

This work was supported in part by the National Natural Science Foundation of China (Grant 62476118), the Natural Science Foundation of Guangdong Province (Grant 2024A1515011759), the Guangdong Science and Technology Program (Grant 2024B1212010002), and the Center for Computational Science and Engineering at Southern University of Science and Technology.

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

# A Inference Time Between Different Methods

We have recorded the inference times for all baseline models as well as HGS, and present them alongside their average optimality gaps in Table 6. For the NCO models, the reported times represent the total duration required to solve a standard test set of 10,000 CVRP100 instances on a single NVIDIA RTX 3090 GPU. It is important to note that HGS runs on a single CPU, and its reported inference time is not directly comparable to the GPU-accelerated times of the learning-based methods.

The results highlight that our proposed method achieves an effective balance between solution quality and computational efficiency. Compared to the baseline model LEHD, our method's inference time is approximately 2.1x longer (57.6s vs. 27.0s). However, this modest computational increase yields a substantial 5.2x reduction in the average optimality gap (from 9.62% to 1.86%). Furthermore, when compared to other recent methods like BQ and INViT, our model is not only more accurate but also faster. For example, our method reduces the optimality gap by over 3.8x compared to BQ (1.86% vs. 7.22%) while being nearly twice as fast (57.6s vs. 109.8s).

Table 6: Total Inference time for solving 10,000 CVRP100 instances. The average inference time (Avg. Time) quantifies a method's average computational efficiency on each dataset (consisting of 10,000 instances).

|           | CVRP100 | |
|-----------|----------|-----------|
|           | Avg. Gap | Avg. Time |
| HGS       | 0.00 %   | 4.5h      |
| AM        | 23.74%   | 2.0s      |
| POMO      | 21.72%   | 3.0s      |
| MDAM      | 16.75%   | 26.8s     |
| BQ        | 7.22%    | 1.8m      |
| LEHD      | 9.62%    | 27.0s     |
| ELG       | 16.29%   | 9.5s      |
| INViT     | 13.28%   | 2.6m      |
| POMO-MTL  | 11.41%   | 3.0s      |
| MVMoE     | 17.24%   | 12.6s     |
| Ours-CVRP | 1.86%    | 57.6s     |

# B Detailed Formulation of Attention Layer

## B.1 Attention Layer

The attention layer [32] is widely used in the neural VRP solver. It comprises a multi-head attention (MHA) block and a node-wise feed-forward (FF) block. Each block incorporates the residual connection [42] and normalization (Norm) [43] operations. Given $X^{(\ell-1)} \in \mathbb{R}^{n \times d}$ and $Y^{(\ell-1)} \in \mathbb{R}^{m \times d}$ as the inputs to the $\ell$-th attention layer, the output $X^{(\ell)}$ is calculated as

$$\hat{X} = \text{Norm}(\text{MHA}(X^{(\ell-1)}, Y^{(\ell-1)}) + X^{(\ell-1)}),$$
$$X^{(\ell)} = \text{Norm}(\text{FF}(\hat{X}) + \hat{X}). \tag{5}$$

For simplicity, the entire process can be represented by

$$X^{(\ell)} = \text{AttnLayer}(X^{(\ell-1)}, Y^{(\ell-1)}). \tag{6}$$

For readability, we omit the $(\ell)$ and $(\ell-1)$ in the following context.

## B.2 Multi-head Attention

We first describe the single-head attention function as follows:

$$\text{Attn}(X, Y) = \text{softmax}\left(\frac{XW_Q(YW_K)^\mathsf{T}}{\sqrt{d}}\right) YW_V, \tag{7}$$

where $X \in \mathbb{R}^{n \times d}$ and $Y \in \mathbb{R}^{m \times d}$ are the input matrices, and $W_Q \in \mathbb{R}^{d \times d_k}, W_K \in \mathbb{R}^{d \times d_k}, W_V \in \mathbb{R}^{d \times d_v}$ are learnable parameters.

The multi-head attention sublayer applies the single-head attention function in Equation (7) for $h$ times in parallel with independent parameters:

$$\text{MHA}(X, Y) = \text{Concat}(\text{head}_1, \ldots, \text{head}_h)W^O,$$
$$\text{head}_i = \text{Attn}_i(X, Y), \tag{8}$$

where for each of $\text{Attn}_i(X, Y)$, $d_k = d_v = d/h$. $W^O \in \mathbb{R}^{d \times d}$ is a learnable matrix.

### B.3 Feed-Forward Layer

$$\text{FF}(X) = \max(0, XW_1 + b_1)W_2 + b_2, \tag{9}$$

where $W_1 \in \mathbb{R}^{d \times d_{ff}}$, $b_1 \in \mathbb{R}^{d_{ff}}$ and $W_2 \in \mathbb{R}^{d_{ff} \times d}$, $b_2 \in \mathbb{R}^d$ are learnable parameters.

## C   Quantification of Problem Similarity

To measure how the similarity of two COPs under different constraint tightness, we propose a problem similarity metric. Intuitively, the similarity between two COPs increases as the difference between their objective values—calculated by applying each problem's optimal solution to the other's constraints—diminishes. Therefore, given two COPs (Problem A and B), we formally define the problem similarity as:

$$\text{Similarity}(A, B) = \underbrace{(1 - \frac{|Obj_B(A) - Obj_B|}{Obj_B})}_{\text{A} \rightarrow \text{B Transferability}} \times \underbrace{(1 - \frac{|Obj_A(B) - Obj_A|}{Obj_A})}_{\text{B} \rightarrow \text{A Transferability}}, \tag{10}$$

where $Obj_A$ and $Obj_B$ are the objective values of the optimal solutions for Problems A and B under their original constraints. $Obj_B(A)$ represents the objective value of Problem A's solution adapted to Problem B's constraints, and $Obj_A(B)$ is defined analogously. The first term quantifies the transferability of Problem A's solution to Problem B, while the second term measures the reverse. High similarity requires both terms to be large.

**Problem Similarity between CVRP and OVRP/TSP**   To validate Equation 10, we compute similarities between three problem classes: TSP, CVRP, and OVRP. We vary the capacity constraint tightness for CVRP instances (CVRP100 dataset) by setting capacities to $C = 10, 50, 400, 500$, ranging from tight to loose. We use the classical solver HGS [21], LKH3 [44], and Concorde [45] to solve the dataset under CVRP, OVRP, and TSP constraints, separately, with results presented in Table 7.

From these results, we observe two key trends: 1) When capacity is small (e.g., $C = 10$), CVRP and OVRP show strong similarity (97.5%) but weak similarity to TSP (25.5%). 2) As capacity increases (e.g., $C = 500$), CVRP's similarity to OVRP drops to 74.1%, while its similarity to TSP rises sharply to 98.2%. This aligns with the observations from Section 4.1, as relaxed capacity constraints make CVRP resemble TSP (a single route with no capacity limits), whereas tight constraints emphasize the routing structure shared with OVRP.

Table 7: Problem similarity analysis across capacity constraints. Transferring rules: OVRP→CVRP adds return trips to the depot; CVRP→TSP removes capacity constraints, and TSP→CVRP reintroduces them. We detail solution transformation in Appendix D.

| | CVRP100 | | | |
| --- | --- | --- | --- | --- |
| | C10 | C50 | C400 | C500 |
| OVRP cost | 30.88 | 9.84 | 7.49 | 7.49 |
| CVRP cost | 59.77 | 15.55 | 8.04 | 7.87 |
| TSP cost | 7.80 | 7.80 | 7.80 | 7.80 |
| OVRP→CVRP cost | 60.67 | 17.25 | 9.78 | 9.74 |
| CVRP→OVRP cost | 31.18 | 11.33 | 7.76 | 7.70 |
| TSP→CVRP cost | 70.47 | 17.91 | 8.78 | 7.98 |
| CVRP→TSP cost | 13.17 | 10.68 | 7.96 | 7.83 |
| Similarity CVRP-OVRP | 97.5% | 75.6% | 75.5% | 74.1% |
| Similarity CVRP-TSP | 25.5% | 53.5% | 88.9% | 98.2% |

Table 8: Analysis of problem similarity between CVRP and CVRPTW under varying time window constraint tightness degrees.

| | CVRP100 | | | |
|---|---|---|---|---|
| $\alpha =$ | 0.2 | 1.0 | 3.0 | 5.0 |
| CVRP cost | 15.51 | 15.51 | 15.51 | 15.51 |
| CVRPTW cost | 33.77 | 24.42 | 15.81 | 15.55 |
| CVRP$\rightarrow$CVRPTW cost | 59.69 | 38.89 | 18.89 | 16.08 |
| CVRPTW$\rightarrow$CVRP cost | 24.42 | 21.24 | 15.80 | 15.55 |
| Similarity CVRP-CVRPTW | 9.9% | 25.7% | 79.0% | 96.3% |

**Problem Similarity between CVRP and CVRPTW**  To validate the versatility of Equation 10, we employ it to quantify the similarity between CVRPTW and CVRP across varying time window constraint tightness degrees. Using the classic heuristic solver HGS [21], we solve CVRP100 instances with vehicle capacity $C = 50$, incorporating time window constraints with tightness coefficients $\alpha \in \{0.2, 1.0, 3.0, 5.0\}$. The results in Table 8 demonstrate a clear trend: under tight constraints (i,e., $\alpha = 0.2$), the similarity drops to 9.9%, with CVRPTW solution costs 2.18 times higher than those of CVRP. In contrast, loose constraints (i,e., $\alpha = 5.0$) yield 96.3% similarity and approximate solution costs. This trend occurs because tight time windows force vehicles to return to the depot frequently after serving only a few customers, resulting in elongated sub-tous. In contrast, under loose time window constraints, vehicles can serve multiple customers before returning, shifting the dominant constraint to capacity, and aligning the behavior more closely with standard CVRP.

## D    Details for Solution Transformation

### D.1    CVRP-TSP

We transform the solutions of CVRP and TSP into each other in the following handcrafted manner.

- **Converting CVRP Solution to TSP Solution**
    1. **Path Segmentation Phase**: We disconnect the edges linking each sub-tour with the depot in the CVRP solution, forming independent path segments. For each isolated node (such as the depot), a virtual path segment consisting of two identical points is constructed.
    2. **Path Reconstruction Phase**: We gradually merge path segments using a greedy strategy. Initially, a random path segment is selected. Then we calculate the length of the selected path between its endpoints and those of the remaining path segments using four connection methods (head-head, head-tail, tail-head, tail-tail) and choose the connection combination with the shortest distance to merge. We iteratively execute this process until all path segments are merged into a single path, and finally close the path into a loop by connecting its ends.

- **Converting TSP Solution to CVRP Solution** Let the depot be any node in the path. The conversion process starts traversing from the starting point of the TSP path, dynamically maintaining the cumulative demand value. When adding the next node would cause the total demand to exceed the vehicle capacity, perform the following operations:
    1. Insert a depot node after the current node.
    2. Reset the cumulative demand value to zero.
    3. Start visiting the next node from the depot again.

    This process continues until the TSP path is fully traversed, ultimately forming a CVRP solution that complies with the capacity constraints.

### D.2    CVRP-OVRP

We transform the solutions of CVRP and OVRP into each other in the following handcrafted manner.

- **Converting CVRP Solution to OVRP Solution** For each sub-tour in the CVRP solution, we first identify the two edges directly connected to the depot, corresponding to the vehicle's

departure path (first edge) and return path (last edge), respectively. We then calculate the Euclidean distances of these two connecting edges and mark the longer one as the final return path. By removing this return path, the closed-loop sub-tour is converted into an open sub-tour, forming a solution that conforms to the characteristics of OVRP.

- **Converting OVRP Solution to CVRP Solution** For each open sub-tour in the OVRP solution, the edge between the end customer node and the depot node is connected, forming a solution that complies with the closed-loop constraints of CVRP.

### D.3 CVRP-CVRPTW

- **Converting CVRP Solution to CVRPTW Solution**

First, we decompose the solution into independent sub-tours. For example, the solution 0-1-2-0-3-4-0 is decomposed into sub-paths [0,1,2,0] and [0,3,4,0], where each sub-tour starts and ends exclusively at the depot.

Second, we perform the time window-based path splitting for each sub-tour:

1. Sequential calculation: Starting from the depot, for each node, we calculate: 1) the arrival time (departure time from the previous node + travel time), 2) the service start time (the maximum value between the arrival time and the node's earliest service time), and 3) the service end time (service start time + service duration).

2. Violation detection: If a node's arrival time exceeds its latest service time, or the return time to the depot exceeds the depot's operating deadline, a new depot node is immediately inserted before this node, splitting the path into two sub-tours.

3. Iterative correction: Repeat the above verification for newly generated sub-tours until all sub-tours satisfy the time window constraints.

Third, the path direction optimization is performed. We perform bidirectional time window-based path splitting for each sub-tour: Calculate the total travel distance of split sub-tours by performing the time window-based path splitting on the sub-tour for both the forward and reverse directions, separately. And finally choose the sub-tour with the shorter distance.

Finally, we collect all the split sub-tours to form a complete CVRPTW solution.

- **Converting CVRPTW solution to CVRP Solution**

1. **Merge optimization phase**: First, we calculate the merge savings value [46] for all sub-tour pairs (the difference between the sum of the distances traveled separately by the two paths and the distance traveled by the merged path) in the CVRPTW solution and establish a priority queue in descending order of savings value. we then sequentially select the top sub-tour pairs from the queue for verification, and perform the merge operation only if the total demand after merging does not exceed the vehicle capacity and the merged sub-tour length is strictly shortened.

2. we repeat the above operation until no more sub-tours can be merged. In this situation, all the merged sub-tours form a legal CVRP solution.

## E  Batch-level vs. Instance-level Strategy for Assigning Constraint Tightness

During model training, the strategy of assigning varying capacities to instances within each batch significantly impacts training outcomes. We compared two capacity allocation schemes: the batch-level random capacity, where all instances in a batch share an identical value randomly sampled from [10, 500] during training, and the instance-level random capacity method, where each instance's capacity is independently sampled from the same interval, allowing heterogeneous capacities within a batch. We train two models under these two settings, with testing results presented in Table 9.

From these results, we observe that the instance-level random capacity strategy consistently outperformed the batch-level one across all datasets. We suspect that this superiority arises from differences in training dynamics: batch-level random capacity causes drastic inter-batch capacity changes, leading to inconsistent parameter update directions and persistent optimization misalignment that hinders gradient descent and convergence quality. In contrast, the instance-level random capacity approach maintains statistically consistent update directions by exposing the model to diverse capacity values within each batch, resulting in stabilized convergence trajectories.

Table 9: Batch-level vs. Instance-level random $C$ for model training.

| | CVRP100 | | | | | | |
| | C10 Gap | C50 Gap | C100 Gap | C200 Gap | C300 Gap | C400 Gap | C500 Gap |
| --- | --- | --- | --- | --- | --- | --- | --- |
| Batch-level random $C$ | 1.99% | 4.66% | 5.02% | 2.46% | 1.06% | 0.89% | 0.94% |
| Instance-level random $C$ | **1.54%** | **3.82%** | **3.67%** | **1.49%** | **0.87%** | **0.75%** | **0.87%** |

## F Quantification of Time Window Constraint Tightness for CVRPTW

**CVRPTW**  Following [47–49], we define the CVRPTW as follows. Building upon CVRP, CVRPTW introduces a time window $[e_i, l_i]$ for each customer node $i$, representing the earliest and latest times at which customer $i$ can be visited. Additionally, each node has a service time $s_i$, which represents the duration a vehicle must remain at the node upon arrival. The depot has a time window $[e_0, l_0]$ but no service time. Each vehicle departs from the depot at time $e_0$ and must return before $l_0$. Let $i$ and $j$ be two consecutive nodes on sub-tour $r$, and let $b_i^r$ denote the time at which the vehicle on sub-tour $r$ arrives at node $i$. The departure time from node $i$ is given by $b_i^r + s_i$, and the arrival time at the next node $j$ is calculated as: $b_j^r = \max(e_j, b_i^r + s_i + t_{ij})$, where $t_{ij}$ is the travel time from node $i$ to node $j$, equal to the Euclidean distance between the two points. The calculation of arrival time indicates that if the vehicle arrives at node $j$ earlier than its start time $e_j$, it must wait until $e_j$ to begin service. If the vehicle arrives at node $j$ later than $l_j$, the sub-tour becomes infeasible. The objective of CVRPTW is to minimize the total Euclidean distance of all sub-tours.

**Time Window Tightness**  Based on the original time windows, we relax or tighten the time window for each customer node as follows:

$$
\begin{aligned}
\delta &= \frac{l_i - e_i}{2} \cdot (1 - \alpha), \\
e_i' &= \max(e_i + \delta, 0) \\
l_i' &= l_i - \delta,
\end{aligned}
\tag{11}
$$

where the original constraints correspond to $\alpha = 1.0$. This formula indicates that a small coefficient represents narrow time windows and a large coefficient represents wide ones. The negative start time will be clipped to zero. In addition, we adjust the service time. Since the service time at each node is already relatively long, increasing it further could lead to the problem becoming infeasible. Therefore, we only allow relaxation of the service time, defined as: $s_i' = \max(\frac{s_i}{\alpha}, s_i)$.

## G Implementation Details of Multi-expert Module in POMO

**POMO model**  The POMO model [9] also has an encoder that generates the node embeddings using the input node features, and a decoder used to convert the node embeddings into the node selection probabilities.

We denote the node embedding matrix generated by the encoder as $H_t = \{h_i^{(L)}\}_{i \in N_a}$, where $N_a$ represents the number of unvisited nodes, $h_i$ is the embedding vector of the $i$-th unvisited node. The context vector is $h_c$, concating the first and last node embeddings. The original decoder is formulated as:

$$
\begin{aligned}
h_c' &= \text{MHA}(h_c, H_t), \\
P &= (C_l \cdot \tanh \frac{(h_c')^\intercal H_t}{\sqrt{d_h}}),
\end{aligned}
\tag{12}
$$

where $C_l$ is a constant.

**POMO model with Multi-expert Module**  For POMO, instead of adding the multi-expert module to the decoder, we add it to the encoder since we empirically find that it works better. The multi-expert module is generally the same as the one presented in Section 4.3.

# H   Discussion on Multi-expert Modules: Discrete Switching and Continuous Generalization

Our multi-expert module, in its current form, functions by switching between specialized experts rather than learning a single, continuously adaptable policy. This is a deliberate design choice informed by empirical investigation.

The goal of learning a unified policy that generalizes continuously across the entire constraint spectrum is indeed highly desirable. We explore this direction by implementing a top-K Mixture-of-Experts (MoE) model with a learnable and soft gating network following MVMoE [1], with $K = 2$. This approach replaces the discrete, rule-based switching with a mechanism that learns to assign weights to different experts, allowing for a weighted combination of their outputs based on the input features. Due to space limitations. We detail the implementation of the top-K MoE module as follows.

**Implementation of Top-K mechanism:**

**1. Gating Mechanism:**

The gating network G is used to select K experts from N experts for each input instance $x$ with a capacity of $C$. Firstly, the gating network calculates a gating score vector based on the instance features $\mathbf{x}$: $S(x) \in \mathbb{R}^N$:

$$S(x) = G(\mathbf{x}).$$

Next, we adopt the Top-K strategy to select the K experts with the highest scores (in our implementation, K = 2). Let $I(x)$ be the index set of the K experts with the highest scores. The final gated weight vector $W(x) \in \mathbb{R}^N$ is obtained by applying the Softmax function to the sparse scores:

$$W_i(x) = \begin{cases} \text{Softmax}(S_i(x)) & \text{if } i \in I(x), \\ 0 & \text{otherwise.} \end{cases}$$

Here, $W_i(x)$ represents the weight assigned to instance $x$ by the $i$-th expert.

**2. Sparse Computation and Output Fusion:**

The key advantage of MoE lies in its Conditional Computation feature. For each instance x, only the K experts selected by the gating network (i.e., $E_i$ for $i \in I(x)$) will be activated and perform the computation. The final output $O(x)$ of the model is the weighted sum of the outputs of these K experts:

$$O(x) = \sum_{i=1}^{N} W_i(x) \cdot E_i(x).$$

Since for the experts with $i \notin I(x)$, $W_i(x) = 0$, this summation is actually only performed on the selected $K$ experts. Thus, while ensuring the model capacity, it significantly saves computational resources.

**3. Load Balancing Auxiliary Loss:**

To prevent the gated network from only activating a few "popular" experts while neglecting the training of other experts, we introduce an auxiliary loss $\mathcal{L}_{aux}$ to encourage load balancing. This loss consists of two parts: the importance loss and the load loss, both of which are achieved by minimizing the square of the coefficient of variation (CV) of the task allocation among experts:

$$\mathcal{L}_{aux} = CV^2(\text{Importance}) + CV^2(\text{Load}).$$

Among them, the "Importance" metric measures the total gating weight that each expert obtains in the batch data B:

$$\text{Importance}_i = \sum_{x \in B} W_i(x).$$

And the "Load" measures the number of times each expert is activated (i.e., selected as the Top-K) in the batch data B. Let $M_i(x)$ be an indicator function, which is 1 when expert $i$ is selected and 0

otherwise:

$$\text{Load}_i = \sum_{x \in B} M_i(x), \quad \text{where} \quad M_i(x) = \begin{cases} 1 & \text{if } i \in I(x), \\ 0 & \text{otherwise.} \end{cases}$$

The square of the coefficient of variation is defined as:

$$CV^2(V) = \left( \frac{\text{std}(V)}{\text{mean}(V) + \epsilon} \right)^2 .$$

Here, $V$ is a vector (such as Importance or Load), and $\epsilon$ is a small constant used to prevent division by zero.

**4. Final Training Objective**

During the training process, we will weigh and sum the main task loss $\mathcal{L}_{main}$ and the auxiliary loss $\mathcal{L}_{aux}$ to form the final optimization objective:

$$\mathcal{L}_{total} = \mathcal{L}_{main} + \alpha \cdot \mathcal{L}_{aux}.$$

Here, $\alpha$ is a hyperparameter used to balance the main task learning and the expert load balancing. In this way, the model can learn a more generalized strategy to handle VRP problems under different constraint intensities. We set $\alpha$ to 0.01 following MVMoE [26].

We train the model with this Top-K MoE module using the same settings as our proposed method (detailed in Section 5) and compare it against the discrete switching mechanism ("Switching"). The models are evaluated on CVRP100 instances with the capacity values $C = \{10, 50, 100, 200, 300, 400, 500\}$. As shown in Table 3, while the Top-K MoE approach is conceptually appealing, the simpler, discrete switching mechanism achieved superior performance across most tightness regimes, resulting in a lower overall average gap (Switching: 1.86% vs. Top-K: 2.17%). Given these empirical results, we select the more performant switching mechanism.

Table 10: Comparison of our proposed discrete switching mechanism against a soft, learnable Top-K MoE gating mechanism on CVRP100 instances.

| | CVRP100 | | | | | | | |
| | C=10 Gap | C=50 Gap | C=100 Gap | C=200 Gap | C=300 Gap | C=400 Gap | C=500 Gap | Varying Capacities Avg. Gap |
| --- | --- | --- | --- | --- | --- | --- | --- | --- |
| Top-K MoE | **1.48%** | 4.58% | 4.49% | 1.94% | 1.07% | 0.77% | **0.83%** | 2.17% |
| Switching | 1.54% | **3.82%** | **3.67%** | **1.49%** | **0.87%** | **0.75%** | 0.87% | **1.86%** |

Furthermore, we would like to emphasize that a primary contribution of this work is the identification and systematic analysis of the constraint tightness-overfitting problem in existing NCO methods. While our proposed solution is straightforward, it effectively alleviates this newly highlighted issue. We believe our findings and methods provide valuable insights that can inspire interesting future works, including the development of more sophisticated, continuously generalized policies.

# I  Discussion on Generality: Extending the Constraint Tightness Framework to More Complex VRP Variants

Our current definitions of "constraint tightness" for capacity and time windows are tailored to the specific VRP variants studied in the paper. This was a deliberate starting point to thoroughly investigate a phenomenon we found to be overlooked.

However, capacity and time windows are two of the most fundamental and ubiquitous constraints in the entire field of vehicle routing. Their presence is standard in a vast array of VRP variants. Therefore, a method that robustly handles the tightness spectrum of these core constraints already possesses significant potential for broad applicability.

To empirically test the universality of our framework, we conduct experiments on more complex VRP variants: the VRP with Backhauls and Time Windows (VRPBTW) and the VRP with Backhauls,

Duration Limit (L), and Time Windows (VRPBLTW). These problems introduce additional structural constraints (backhaul and duration limit) on top of the standard capacity and time window constraints. We apply our proposed method (VCT training and the MEM module) to the time window constraint in these new, more complex settings, using POMO as the base model. We follow MVMoE [26] to configure the settings of additional constraints, backhaul, and duration limit.

The results presented in Table 11 demonstrate the effectiveness of our approach. From the table, we can observe that the standard POMO model once again suffers from severe performance degradation as the time window constraint moves to be either tighter or looser than the standard $\alpha = 1.0$. In contrast, our enhanced model (POMO + our method) significantly mitigates this issue, maintaining much more stable and superior performance across the entire tightness spectrum.

These results strongly suggest that our framework is not a fragile solution limited to simple problems. Instead, it can function as a robust and adaptable sub-system. It can be integrated into solvers for more complex problems to specifically handle the challenge of varying constraint tightness, without conflicting with the logic required for other constraints. We hope this work brings attention to the critical issue of constraint tightness in the NCO community and inspires future research to develop diverse strategies for this challenge across a wider range of combinatorial optimization problems.

Table 11: Performance on Complex VRP Variants (i.e., VRPBTW and VRPBLTW) with Varying Time Window Tightness

| | VRPBTW100 | | | | | | | |
| | alpha=0.2 Gap | alpha=0.5 Gap | alpha=1 Gap | alpha=1.5 Gap | alpha=2 Gap | alpha=2.5 Gap | alpha=3 Gap | Varying Time Windows Avg. Gap |
|---|---|---|---|---|---|---|---|---|
| OR-Tools | 0.00% | 0.00% | 0.00% | 0.00% | 0.00% | 0.00% | 0.00% | 0.00% |
| POMO | 18.81% | 13.35% | **8.46%** | 12.12% | 19.42% | 30.27% | 43.96% | 20.91% |
| POMO + Our Method | **13.23%** | **11.97%** | 9.34% | **8.67%** | **9.07%** | **8.80%** | **9.11%** | **10.03%** |

| | VRPBLTW100 | | | | | | | |
| | alpha=0.2 Gap | alpha=0.5 Gap | alpha=1 Gap | alpha=1.5 Gap | alpha=2 Gap | alpha=2.5 Gap | alpha=3 Gap | Varying Time Windows Avg. Gap |
|---|---|---|---|---|---|---|---|---|
| OR-Tools | 0.00% | 0.00% | 0.00% | 0.00% | 0.00% | 0.00% | 0.00% | 0.00% |
| POMO | 18.90% | 13.32% | **8.29%** | 11.42% | 17.06% | 23.73% | 31.16% | 17.70% |
| POMO + Our Method | **13.00%** | **11.80%** | 9.09% | **8.42%** | **8.92%** | **8.45%** | **8.56%** | **9.75%** |

## J Formal Definition of Constraint Tightness

A more rigorous and general definition of "constraint tightness" is important, and a more general definition that is independent of specific problem features is shown as follows.

We propose to define the "**tightness**" of a constraint as: **the degree of degradation in the objective value of a solution produced by a reference algorithm due to the introduction of that constraint.**

The intuition is that tighter constraints restrict the solution space, forcing a reference algorithm to find higher-cost solutions. That is, a constraint acts as an impediment to the heuristic algorithm, forcing it to construct a less direct, more complex solution path. This manifests as a quantifiable increase in the final objective value. For example, in CVRP, a tight capacity constraint (e.g., $C = 10$) necessitates frequent returns to the depot, resulting in many short, high-cost sub-tours. This contrasts with a loose constraint (e.g., $C = 500$), where the solution structure resembles a single, efficient TSP tour.

In this paper, we use simple, general heuristics like Nearest Neighbor (NN) as reference algorithms. We avoid using strong solvers to ensure that our definitions are both practically simple and general. Furthermore, simple heuristics are computationally inexpensive, facilitating an efficient analysis.

**1. Formal Definition of Constraint Tightness**

We formalize this definition with the following equation. Let $P$ be a combinatorial optimization (CO) problem instance, with the objective of minimizing the function $f(\pi)$, where $\pi$ is a solution to the instance:

- Let $P_\xi$ be the instance with a specific constraint $\xi$.
- Let $P_\emptyset$ be the unconstrained baseline, representing the problem with constraint $\xi$ fully relaxed.

- Let $P_1$ be the extreme-constraint version, where the constraint $\xi$ is exceedingly tight, serving as an upper bound for the solution objective value degradation.

- Let $\mathcal{H}$ be a simple, general deterministic heuristic solving algorithm. It serves as the reference algorithm.

- The heuristic $\mathcal{H}$ can find a feasible solution to the problem $P$.

- Let $f(\mathcal{H}(P))$ be the objective value of the solution generated by the algorithm $\mathcal{H}$ for an instance $P$.

- The value of $f$ used to compute tightness refers to the objective value of a feasible solution.

- The range of the function $f$ is strictly greater than 0.

Then, the tightness $T(\xi)$ of the constraint $\xi$ can be defined as:

$$T(\xi, P, \mathcal{H}) = \frac{log(f\left(\mathcal{H}\left(P_\xi\right)\right)) - log(f\left(\mathcal{H}\left(P_\emptyset\right)\right))}{log(f\left(\mathcal{H}\left(P_1\right)\right)) - log(f\left(\mathcal{H}\left(P_\emptyset\right)\right))} \tag{13}$$

**Interpretation of the Formula:**

- **This formula calculates a normalized rate of the objective value degradation caused by constraint $\xi$.** The numerator measures the logarithmic increase in the objective value relative to the unconstrained baseline $P_\emptyset$. This is then normalized by the denominator, which measures the maximum possible logarithmic increase in the objective value from the "unconstrained" to the "extreme" case.

- The resulting tightness score, $T(\xi)$, **is a dimensionless value within the range [0,1]**. A score of $T(\xi) = 0$ signifies a completely "loose" constraint with no negative impact on the solution. $T(\xi) = 1$ indicates that the constraint imposes the maximum possible solution objective value degradation, equivalent to the extreme case.

- **Logarithmic Scaling:** We observe that objective function values in CO problems can change sharply as the constraint tightness changes (as shown in the table below). Therefore, we employ the logarithm to compress this scale, making the tightness metric more stable.

**This formula offers several key advantages:**

1. **Generality:** It quantifies tightness by observing the impact of the constraint on the solution generated by a general heuristic algorithm, rather than relying on problem-specific physical parameters (such as capacity, time window width, etc.). It is therefore applicable to any combinatorial optimization problem for which "unconstrained" and "extreme" baseline cases can be defined.

2. **Absolute Quantifiability:** It provides a dimensionless scalar value for any given instance, enabling meaningful tightness comparisons across different instances.

3. **Reproducibility**: By using the simple, general, and deterministic heuristic (e.g., Nearest Neighbor) as a reference algorithm, the tightness calculation is fully reproducible and computationally inexpensive.

While developing this definition, we also considered alternative definitions, such as those based on solution structure (e.g., number of sub-tours) or constraint features (e.g., capacity values). These were found to be either too problem-specific or difficult to generalize. We therefore selected defining tightness via its impact on the objective function value, as mentioned above.

**2. Generalization to Different Problems**

To demonstrate the versatility of this general definition, we apply it to three CO problems: two VRP variants (CVRP and CVRPTW) and the classic Knapsack Problem (KP) as follows.

**2.1 For CVRP:**

- Constrained Instance ($P_\xi$): A standard CVRP instance with a finite vehicle capacity $C_{cap}$.

- Unconstrained Baseline ($P_\emptyset$): This baseline is formed by relaxing the capacity value to a sufficiently large value to serve all customers in a single tour.

- Extreme Constraint Baseline ($P_1$): This baseline represents the most restrictive scenario, where constraints are so tight (e.g., capacity is minimal) that each vehicle can serve only one customer before returning to the depot.

- Reference Algorithm ($\mathcal{H}$): We use a Nearest Neighbor (NN) heuristic adapted for the CVRP. The algorithm iteratively constructs routes by adding the closest unvisited customer. If adding a customer would violate the vehicle's remaining capacity, the current route is finalized, and a new route begins from the depot.

## 2.2 For CVRPTW:

- Constrained Instance ($P_\xi$): A standard CVRPTW instance with time window constraints.

- Unconstrained Baseline ($P_\emptyset$): This baseline is derived by relaxing the time windows to be infinitely large.

- Extreme Constraint Baseline ($P_1$): Similar to the CVRP case, extremely tight time windows can force a vehicle to return to the depot after each delivery, resulting in single-customer routes.

- Reference Algorithm ($\mathcal{H}$): The reference heuristic is an adapted Nearest Neighbor (NN) algorithm. It extends the CVRP's NN logic by verifying both capacity and time window feasibility before appending a customer to a route.

## 2.3 For the Knapsack Problem (KP):

- Problem Description: Given a set of items, each with a weight and a value randomly sampled from (0, 1), select which items to include in a knapsack so that the total weight is less than or equal to a given limit weight capacity $C_{kp}$. The objective is to maximize the total value of items in the knapsack. The instances used in this study, denoted as KP200, consist of 200 items.

- Constrained Instance ($P_\xi$): A standard KP instance with a knapsack capacity constraint $C_{kp}$.

- Unconstrained Baseline ($P_\emptyset$): This corresponds to a knapsack with infinite capacity. The optimal solution is trivial: select all items.

- Extreme Constraint Baseline ($P_1$): This occurs when the knapsack capacity is set to the minimum weight among all available items. Consequently, the only feasible non-empty solutions involve selecting the single, lightest item.

- Reference Algorithm ($\mathcal{H}$): The reference is a standard greedy algorithm. It sorts items by their value-to-weight ratio in descending order and sequentially adds them to the knapsack until no more items can fit within the capacity limit.

## 3. Experimental Validation

We evaluate the proposed tightness formula across the CVRP, CVRPTW, and KP instances. The results, summarized in Table 1, reveal a clear and consistent trend: as constraints are relaxed from the extreme case towards the unconstrained baseline, the calculated **Tightness** value smoothly transitions from 1 to 0. For example, in the CVRP instances, increasing the vehicle capacity ($C_{cap}$) from 10 to 50 and then to 500 causes the tightness to decrease from 0.837 to 0.280 and 0.018, respectively, bridging the gap between the extreme case (Tightness = 1.0) and the unconstrained TSP (Tightness = 0.0).

These findings confirm that our metric provides a robust and intuitive measure of constraint intensity. It successfully quantifies the absolute tightness of a single instance while also enabling direct comparison of relative tightness between different instances.

In addition, we would like to clarify that the analysis in our main paper explores a comprehensive range of tightness values, i.e., [0.018, 0.837] for CVRP with $C \in [10, 500]$ and [0.072, 0.717] for CVRPTW with $\alpha \in [0.2, 3]$. This scope is broad enough to reflect the overfitting issue associated with the NCO model trained on CVRP100 (tightness = 0.280) and CVRPTW (tightness = 0.418).

Table 12: Tightness Value Calculation Results for Different Problems under Various Degrees of Constraint Tightness. "Extreme Case" means that the constraint is exceedingly tight, serving as an upper bound for the solution objective value degradation.

| Method | | CVRP100 Extreme Case | C=10 | C=50 | C=100 | C=200 | C=300 | C=400 | C=500 | Unlimited C |
|---|---|---|---|---|---|---|---|---|---|---|
| NN | Obj. Value ↓ | 103.986 | 70.684 | 18.886 | 13.774 | 11.329 | 10.415 | 10.494 | 10.146 | 9.726 |
| | Tightness | 1.000 | 0.837 | 0.280 | 0.147 | 0.064 | 0.029 | 0.032 | 0.018 | 0.000 |
| | | CVRPTW100 Extreme Case | $\alpha$=0.2 | $\alpha$=0.5 | $\alpha$=1 | $\alpha$=1.5 | $\alpha$=2 | $\alpha$=2.5 | $\alpha$=3 | Unlimited $\alpha$ |
| NN | Obj. Value ↓ | 103.986 | 64.422 | 51.709 | 40.473 | 30.436 | 25.687 | 23.043 | 21.611 | 19.136 |
| | Tightness | 1.000 | 0.717 | 0.587 | 0.443 | 0.274 | 0.174 | 0.110 | 0.072 | 0.000 |
| | | KP200 | | | | | | | | |
| | | Extreme Case | C=0.1 | C=1 | C=2 | C=5 | C=10 | C=25 | C=50 | Unlimited C |
| Greedy | Obj. Value ↑ | 0.532 | 3.346 | 11.217 | 16.103 | 25.743 | 36.477 | 57.655 | 81.116 | 99.865 |
| | Tightness | 1.000 | 0.649 | 0.418 | 0.349 | 0.259 | 0.192 | 0.105 | 0.040 | 0.000 |

# K  Additional Experimental Demonstration on the Model's Overfitting to Capacity Constraints

Analyzing how the solution structure changes with varying capacity provides more direct and powerful evidence for our claim that existing models overfit to the capacity constraint on which they are trained. To provide this justification, we conduct a supplementary analysis focusing on a key aspect of the solution structure: the average number of vehicles used. We compare the solutions generated by three representative neural solvers (POMO, BQ, and LEHD), all pre-trained on CVRP100 instances with C=50, against the (near)-optimal solutions from the classical HGS solver. The evaluation is performed on datasets with varying capacities (C = 10, 50, 100, 500), each containing 10,000 instances. The results presented in Table 13 show the average number of vehicles used by each method and the percentage gap relative to the HGS baseline.

Table 13: Comparison of average number of vehicles used across different capacities

| Method | C=10 Avg. Vehicle Num. | Vehicle Num. Gap | C=50 Avg. Vehicle Num | Vehicle Num. Gap | C=100 Avg. Vehicle Num | Vehicle Num. Gap | C=500 Avg. Vehicle Num | Vehicle Num. Gap |
|---|---|---|---|---|---|---|---|---|
| HGS | 53.19 | 0.00% | 10.44 | 0.00% | 5.44 | 0.00% | 1.44 | 0.00% |
| POMO | 57.18 | 7.50% | 10.66 | **2.11%** | 6.21 | 14.15% | 3.73 | 159.03% |
| BQ | 56.84 | 6.86% | 10.62 | **1.72%** | 5.61 | 3.13% | 1.91 | 32.64% |
| LEHD | 65.43 | 23.01% | 10.95 | **4.89%** | 5.86 | 7.72% | 1.94 | 34.72% |

From these results, we can observe that the gap in the average number of vehicles used by the neural models, relative to the HGS baseline, is minimized at C=50. As the capacity deviates from this training value, i.e., becoming either tighter (C=10) or looser (C=100, 500), the gap increases substantially. For example, the gap for LEHD at C=10 and C=500 is 4.7× and 7.1× larger, respectively, than its gap at C=50. This demonstrates that when faced with instances requiring a fundamentally different solution structure (e.g., ≈53 vehicles for C=10 or ≈1 vehicles for C=500), the models fail to adapt. They generate solutions with a suboptimal number of vehicles, which directly contributes to the higher overall travel costs reported in our main paper.

# L  Explanation and rationality analysis regarding the selection of the constraint range $[C_{min}, C_{max}]$

We would like to offer a detailed discussion about how to determine this range in practical applications.

**1. Determining the Range in Practical Applications**

In practical applications, a direct and effective method is to analyze historical operational data. For example, a logistics company could derive an empirical range that reflects its specific business characteristics by statistically analyzing the vehicle capacities and total customer demands from its past tasks. Our method can be flexibly applied to train a model on any such empirically derived range $[C_{min}, C_{max}]$ to achieve promising performance for that specific operational context.

**2. Principled Rationale for the Range Selection in Our Paper.**

The primary motivation for selecting the range [10, 500] is to comprehensively cover the entire spectrum of the CVRP100 problem's characteristics within a controlled experimental environment. **(a) Lower Bound** $C_{min} = 10$**:** This value is set to simulate a tight constraint scenario. In our experimental setup, following the seminal works [18, 9], customer demands are uniformly distributed in $\{1, ..., 9\}$. A vehicle capacity of 10 means a vehicle can serve only two customers on average, pushing the problem to the brink of feasibility. **(b) Upper Bound** $C_{max} = 500$**:** This value is set to simulate a loose constraint scenario. For a CVRP100 instance with 100 customers and an average demand of 5, the total expected demand is 500. When the capacity approaches this value, the capacity constraint becomes nearly irrelevant. This range ensures that our model is exposed to the full spectrum of problem structures during training. This is crucial for thoroughly analyzing and addressing the overfitting issue of NCO models under varying constraints.

**3. Representativeness of the Chosen Range on Real-World Benchmarks**

To further validate that our chosen range holds practical relevance, we analyze 192 instances from the classic CVRPLIB benchmarks (A, B, E, F, M, P, and X sets). We find that across these real-world instances, the number of customers served by a single vehicle in a sub-tour ranges from a minimum of 2 to a maximum of 27. In our experimental setting, a vehicle can serve between 2 (for $C = 10$) and 100 (for $C = 500$) customers. This demonstrates that our training setup's range of possible sub-tour sizes fully encompasses those observed in these widely used real-world benchmarks. This strong quantitative evidence shows that our selected range is not only conceptually comprehensive but also highly representative of real-world application scenarios.

## M Analysis of the Computational Cost of the Multi-Expert Module

We record the average inference time and optimality gap for our model with and without the module, with the results presented in Table 14 below. The results clearly demonstrate that the MEM module achieves a significant improvement in solution quality at a modest computational cost.

Specifically, incorporating the MEM reduces the average optimality gap from 2.41% to 1.86%, which constitutes a relative performance improvement of 23%. We acknowledge that this performance gain is accompanied by a moderate increase in inference time (from 0.67 min to 0.96 min). However, we believe that this trade-off is highly favorable. In many real-world application scenarios (e.g., logistics planning and resource scheduling), the economic benefits derived from a higher-quality solution (e.g., reduced travel distance and vehicle costs) typically far outweigh this one-time computational expense.

Table 14: Impact of the Multi-Expert Module (MEM) on Performance and Inference Time. Both models are trained with varying constraint tightness.

| | CVRP100 Varying Capacities | |
| --- | --- | --- |
| | Avg. Gap | Avg. Time |
| Ours without MEM | 2.41% | 0.67m |
| Ours with MEM | **1.86%** | 0.96m |

## N Impact of Node Demand Distribution on Model Performance

The distribution of node demands is also a critical factor influencing model performance. While we did consider this factor in our initial experimental design, we made a deliberate choice to isolate a single variable, vehicle capacity, for our study. This approach allows us to clearly and rigorously demonstrate the constraint overfitting issue inherent in existing methods.

A heterogeneous demand distribution would likely pose an even greater challenge. We have conducted an experiment to validate this. We create a new test dataset where customer demands follow a more realistic heterogeneous distribution: a long-tailed distribution. We choose this distribution because, in many real-world applications (such as e-commerce and parcel delivery), the majority of customers have small demands, while only a few have very large demands.

Specifically, we generate the long-tailed demand distribution as follows: for each possible demand value $d \in D = \{1, 2, 3, 4, 5, 6, 7, 8, 9\}$, its probability of occurrence $P(d)$ is given by the formula below:

$$P(d) = \frac{\frac{1}{d^\delta}}{\sum_{i=1}^{9} \frac{1}{i^\delta}},$$

where the hyperparameter $d = 1.5$ controls the skewness of the distribution. The specific probabilities for each demand value are shown in Table 15:

Table 15: Probabilities of demand values under a long-tailed distribution.

| demand (d) = | 1 | 2 | 3 | 4 | 5 | 6 | 7 | 8 | 9 |
|---|---|---|---|---|---|---|---|---|---|
| P(d) = | 0.5092 | 0.1800 | 0.098 | 0.0637 | 0.0455 | 0.0346 | 0.0275 | 0.0225 | 0.0189 |

On this basis, we evaluate the performance of several NCO models across different scenarios. These models are pre-trained on instances with a uniform demand distribution (C=50, CVRP100). The test scenarios include: **(a) Fixed Capacity (C=50):** Tested under both uniform and long-tailed demand distributions. **(b) Varying Capacities:** Capacity $C$ is drawn from the set $\{10, 50, 100, 200, 300, 400, 500\}$, also tested under both uniform and long-tailed demand distributions. For each of these seven capacity values, the corresponding test set comprises 10,000 instances. The experimental results are shown in Table 16.

Table 16: Impact of demand distribution on exisiting models' performance. The bolded Gap indicates a significant decline in performance of the model when the distribution of demand changes to a Long-Tailed pattern.

| | CVRP100 | | | |
|---|---|---|---|---|
| | C=50 | | Varying Capacities | |
| Method | Uniformed Demand Gap | Long-Tailed Demand Gap | Uniformed Demand Avg. Gap | Long-Tailed Demand Avg. Gap |
| HGS | 0.00% | 0.00% | 0.00% | 0.00% |
| AM | 7.66% | **14.10%** | 23.76% | 23.58% |
| POMO | 3.66% | **12.16%** | 21.72% | **26.04%** |
| MDAM | 5.38% | **13.71%** | 16.75% | **19.41%** |
| BQ | 3.25% | **3.53%** | 7.22% | **7.39%** |
| LEHD | 4.22% | **4.54%** | 9.62% | 7.98% |
| ELG | 5.24% | **11.29%** | 16.29% | **18.92%** |
| INViT | 7.85% | **12.31%** | 13.28% | **13.87%** |
| POMO-MTL | 4.50% | **6.93%** | 11.41% | **12.76%** |
| MVMoE | 5.06% | **8.53%** | 17.24% | **19.29%** |

The results clearly show that under the fixed capacity (C=50) setting, when the demand distribution shifts from uniform to long-tailed, almost all evaluated NCO models experience a significant performance degradation. For instance, POMO's gap drastically increases from 3.66% to 12.16%. Under the varying capacities setting, most models also exhibit a trend of performance decline. These experimental results strongly confirm your assessment: the distribution of customer demands is indeed a crucial factor affecting the generalization capability of NCO models. The performance of existing models is substantially compromised under more realistic and challenging heterogeneous demand distributions.

# O Generalization of the Constraint Overfitting Issue: From VRPs to Other CO Problems

The overfitting issue with respect to constraint tightness, which we discuss in the context of VRPs, also manifests in other CO problems. It can be observed in other CO problems, such as the Knapsack Problem (KP) with its capacity limit, the Bin Packing Problem (BPP) with bin capacities, or the Job-Shop Scheduling Problem (JSSP) with task completion deadlines.

To provide empirical evidence, we conduct a case study on KP. Following the setup of POMO, we generated KP200 instances with varying capacities: $C \in \{2, 5, 10, 25, 50\}$. We then evaluate two representative models, POMO and BQ, using their publicly available checkpoints that are trained specifically on KP200 instances with a fixed capacity of C=25. Both models used greedy search for inference. The results are presented in Table 17.

Table 17: Performance of baseline NCO models on KP200 instances with different capacities. The larger the value is, the better the model performs.

| Method | KP200 | | | | | | | | | |
| | C=2 | | C=5 | | C=10 | | C=25 | | C=50 | |
| | Value | Gap | Value | Gap | Value | Gap | Value | Gap | Value | Gap |
| OR-Tools | 16.157 | 0.000% | 25.723 | 0.000% | 36.467 | 0.000% | 57.665 | 0.000% | 81.163 | 0.000% |
| POMO | 9.673 | 40.132% | 23.539 | 8.488% | 34.428 | 5.590% | 57.393 | **0.472%** | 79.855 | 1.611% |
| BQ | 15.653 | 3.119% | 25.546 | 0.686% | 36.369 | 0.269% | 57.611 | **0.094%** | 80.624 | 0.664% |

As shown in the table, the performance of both POMO and BQ degrades significantly as the capacity C deviates from the training value of 25. For the tight constraint C=2, the POMO model's gap explodes to 40.132%, an 85× increase compared to its performance at C=25. Similarly, the BQ model's gap at C=2 is 33× larger than its gap at C=25. When the constraint becomes looser at C=50, the performance drop is also evident, with the gaps for POMO and BQ increasing by 3.4× and 7×, respectively. These experimental results on KP strongly demonstrate that the constraint overfitting issue also occurs in other CO problems extending beyond VRPs.

# P Licenses

The licenses for the codes used in this work are listed in Table 18.

Table 18: Licenses for codes used in this work

| Resource | Type | Link | License |
|---|---|---|---|
| LKH3 [44] | Code | http://webhotel4.ruc.dk/ keld/research/LKH-3/ | Available for academic research use |
| HGS [21] | Code | https://github.com/chkwon/PyHygese | MIT License |
| Concorde [50] | Code | https://github.com/jvkersch/pyconcorde | BSD 3-Clause License |
| POMO [9] | Code | https://github.com/yd-kwon/POMO | MIT License |
| LEHD [20] | Code | https://github.com/CIAM-Group/NCO_code/tree/main/single_objective/LEHD | MIT License |
| BQ [19] | Code | https://github.com/naver/bq-nco | CC BY-NC-SA 4.0 |
| INViT [24] | Code | https://github.com/Kasumigaoka-Utaha/INViT | Available for academic research use |
| ELG [23] | Code | https://github.com/gaocrr/ELG | MIT License |
| POMO-MTL [25] | Code | https://github.com/FeiLiu36/MTNCO | MIT License |
| AM [18] | Code | https://github.com/wouterkool/attention-learn-to-route | MIT License |
| MDAM [22] | Code | https://github.com/liangxinedu/MDAM | MIT License |

