# OpenReview forum: "Rethinking Neural Combinatorial Optimization for Vehicle Routing Problems with Different Constraint Tightness Degrees"
_NeurIPS.cc/2025/Conference — NeurIPS 2025 poster_

### Official Review · Reviewer_acR7 · 2025-06-27

**Clarity:** 3
**Significance:** 3
**Originality:** 2
**Rating:** 5
**Confidence:** 4

**Summary:**

This paper systematically identifies and addresses a critical limitation in existing NCO models: overfitting to fixed constraint tightness. The proposed multi-expert module technique is sound, empirically effective, and practically applicable. Despite some minor shortcomings (e.g., lack of efficiency analysis), the work is of high quality and makes a meaningful contribution to the NCO literature.

**Questions:**

1.	Have you considered the impact of the node demand distribution? Most neural methods simply employ uniform node demands, which means that the capacity constraint is just counting the number of visited nodes. If the node demands were drawn from a more heterogeneous distribution, would the performance of current methods be similarly affected or further compromised?

2.	Does the overfitting issue discussed in the context of VRPs also manifest in other combinatorial optimization problems?

**Ethical Concerns:**

["NO or VERY MINOR ethics concerns only"]

**Final Justification:**

This paper is the first to reveal the overfitting issue of neural methods with respect to constraint tightness (e.g., vehicle capacity and time window tightness), and proposes well-motivated solutions that empirically mitigate this problem. This contribution is both significant and timely for the field of neural combinatorial optimization.

My earlier concerns primarily pertained to the generalizability of the conclusions. The authors' rebuttal convincingly demonstrates that their findings extend to other constraint-related features (e.g., node demand) and to other constrained combinatorial optimization problems beyond the VRP, which has effectively addressed my concerns.

**Limitations:**

Yes

**Quality:**

3

**Strengths And Weaknesses:**

Strengths:
1. This paper, for the first time, reveals that many NCO methods tend to overfit a certain degree of constraint tightness, which is a critical yet underexplored limitation. This insight represents a meaningful and timely contribution to the field.

2. The proposed multi-expert module is conceptually simple yet empirically effective, significantly enhancing the model's capacity to handle varying levels of constraint tightness. Furthermore, the authors demonstrate that this modular design can be seamlessly integrated into a broad range of backbone architectures, indicating its versatility and general applicability.

3. The paper provides a comprehensive empirical evaluation, including experiments across multiple models, varying constraint levels, and different problem variants. In addition, the inclusion of ablation studies and problem similarity analyses offers strong empirical support for the paper’s central claims.

Weaknesses:
1. The proposed training approach assumes prior knowledge of the constraint tightness range $[C_{min}, C_{max}]$, but there is little discussion about how to determine this range in practical applications.

2. While the proposed multi-expert module improves performance, its additional computational cost is not analyzed or quantified.

---

> ### Author Rebuttal · Authors · 2025-07-31
>
> Dear Reviewer acR7,
>
> Thank you very much for taking the time and effort to review our work. We are delighted that you find that our paper is the first to reveal the critical yet underexplored limitation that NCO methods tend to overfit a certain degree of constraint tightness, representing a meaningful and timely contribution, and that our comprehensive empirical evaluation provides strong support for the paper's central claims.
>
> We address your concerns point-by-point as follows.
>
> > **W1. There is little discussion about how to determine the constraint tightness range [$C_{min}$, $C_{max}$] in practical applications.**
>
> Thank you very much for your valuable comment. We would like to provide a detailed discussion about how to determine this range in practical applications.
>
> **1. Determining the Range in Practical Applications**
>
> In practical applications, a direct and effective method is to analyze historical operational data. For example, a logistics company could derive an empirical range that reflects its specific business characteristics by statistically analyzing the vehicle capacities and total customer demands from its past tasks.
>
> **2. Rationale for the Range Selection in Our Paper.**
>
> The primary motivation for selecting the range [10, 500] is to comprehensively cover the entire spectrum of the CVRP100 problem's characteristics. **(a) Lower Bound $C_{min}=10$:** This value is set to simulate a tight constraint scenario. In our experimental setup, customer demands are uniformly distributed in {1, ..., 9}$. C=10 means a vehicle can serve only two customers on average, pushing the problem to the brink of feasibility. **(b) Upper Bound $C_{max}=500$:** This value is set to simulate a loose constraint scenario. For a CVRP100 instance with 100 customers and an average demand of 5, the total expected demand is 500. When the capacity approaches this value, the capacity constraint becomes nearly irrelevant. This range is crucial for thoroughly analyzing and addressing the overfitting issue of NCO models under varying constraints.
>
> **3. Representativeness of the Chosen Range on Real-World Benchmarks**
>
> To validate that our chosen range holds practical relevance, we analyze 192 instances from the classic CVRPLIB benchmarks (A, B, E, F, M, P, and X sets). We find that across these real-world instances, the number of customers served by a single vehicle ranges from a minimum of 2 to a maximum of 27 on average. In our experimental setting, a vehicle can serve between 2 (for $C=10$) and 100 (for $C=500$) customers. This quantitative evidence shows that our selected range is not only conceptually comprehensive but also highly representative of real-world application scenarios.
>
>
>
> > **W2. While the proposed multi-expert module improves performance, its additional computational cost is not analyzed or quantified.**
>
> Thank you very much for this constructive comment. To quantify the additional computational cost of the multi-expert module (MEM), we record the inference time and optimality gap for our model with and without the module, with the results presented in Table 7 below.
> From these results, we can observe that incorporating the MEM reduces the average optimality gap from 2.41\% to 1.86\%, which constitutes a relative performance improvement of 23\%. We acknowledge that this performance gain is accompanied by a moderate increase in inference time (from 0.67 min to 0.96 min). However, we believe that this trade-off is acceptable. As shown in Table 8, when compared to two recent methods like BQ and INViT, our model is not only more accurate but also faster.
>
> Table 7: Impact of the Multi-Expert Module (MEM) on Performance and Inference Time.
> ||CVRP100 （Varying Capacities）||
> |-|:-:|:-:|
> ||Avg. Gap|Avg. Time|
> |Ours without MEM|2.41\%|0.67m|
> |Ours with MEM|**1.86\%**|0.96m|
>
>
>
> Table 8: Total Inference time of baseline models for solving 10,000 CVRP100 instances.
> |  | CVRP100 （Varying Capacities） |  |
> |---|:---:|:---:|
> |  | Avg. Gap | Avg. Time |
> | BQ | 7.22\% | 1.8m |
> | INViT | 13.28\% | 2.6m |
> | Ours | **1.86\%** | 0.96m |
>
>
> > **Q1. Have you considered the impact of the node demand distribution? If the node demands were drawn from a more heterogeneous distribution, would the performance of current methods be similarly affected or further compromised?**
>
> Thank you very much for raising this insightful question. While we did consider the factor of the distribution of node demands in our initial experimental design, we made a deliberate choice to isolate a single variable—vehicle capacity—for our study. This approach allows us to clearly and rigorously demonstrate the constraint overfitting issue inherent in existing methods.
>
> A heterogeneous demand distribution would likely pose an even greater challenge. We conduct an experiment to validate this. We generate a new test dataset where customer demands follow a more realistic heterogeneous distribution: a long-tailed distribution. We choose this distribution because, in many real-world applications (such as e-commerce and parcel delivery), the majority of customers have small demands, while only a few have very large demands. Specifically, in the long-tailed distribution, each possible demand value $d\in$ {1, 2, 3, 4, 5, 6, 7, 8, 9}, and its probability of occurrence $P(d)$ is given by the formula below:
>
> $$P(d) = \frac{\frac{1}{d^\delta}}{\sum_{i=1}^{9} \frac{1}{i^\delta}}$$
>
> where the hyperparameter $d=1.5$ controls the skewness of the distribution. The specific probabilities for each demand value are shown in Table 9:
>
> Table 9: Probabilities of demand values under a long-tailed distribution
> |demand(d)=|1|2|3|4|5|6|7|8|9|
> |-|:-:|:-:|:-:|:-:|:-:|:-:|:-:|:-:|:-:|
> |P(d)=|0.5092|0.1800|0.098|0.0637|0.0455|0.0346|0.0275|0.0225|0.0189|
>
>
> On this basis, we evaluate the performance of several NCO models across different scenarios. These models are pre-trained on instances with a uniform demand distribution (C=50, CVRP100). The test scenarios include: **(a) Fixed Capacity (C=50):** Tested under both uniform and long-tailed demand distributions. **(b) Varying Capacities:** Capacity $C$ is drawn from the set {10, 50, 100, 200, 300, 400, 500}, also tested under both uniform and long-tailed demand distributions. The experimental results are shown in Table 10.
>
> Table 10: Impact of demand distribution on existing models' performance.
> ||CVRP100||||
> |:-:|:-:|:-:|:-:|:-:|
> ||C=50|C=50|Varying Capacities|Varying Capacities|
> ||Uniformed Demand|Long-Tailed Demand|Uniformed Demand|Long-Tailed Demand|
> |Method|Gap|Gap|Avg.Gap|Avg. Gap|
> |HGS|0.00\%|0.00\%|0.00\%|0.00\%|
> |AM|**7.66\%**|14.10\%|23.76\%|**23.58\%**|
> |POMO|**3.66\%**|12.16\%|**21.72\%**|26.04\%|
> |MDAM|**5.38\%**|13.71\%|**16.75\%**|19.41\%|
> |BQ|**3.25\%**|3.53\%|**7.22\%**|7.39\%|
> |LEHD|**4.22\%**|4.54\%|9.62\%|**7.98\%**|
> |ELG|**5.24\%**|11.29\%|**16.29\%**|18.92\%|
> |INViT|**7.85\%**|12.31\%|**13.28\%**|13.87\%|
> |POMO-MTL|**4.50\%**|6.93\%|**11.41\%**|12.76\%|
> |MVMoE|**5.06\%**|8.53\%|**17.24\%**|19.29\%|
>
>
> The results clearly show that under the fixed capacity (C=50) setting, when the demand distribution shifts from uniform to long-tailed, almost all evaluated NCO models experience a significant performance degradation. For instance, POMO's gap drastically increases from 3.66\% to 12.16\%. Under the varying capacities setting, most models also exhibit a trend of performance decline. These experimental results strongly confirm that the performance of existing models is substantially compromised under more realistic and challenging heterogeneous demand distributions.
>
>
> > **Q2. Does the overfitting issue discussed in the context of VRPs also manifest in other combinatorial optimization problems?**
>
> Thank you very much for this insightful question. The overfitting issue with respect to constraint tightness, which we discuss in the context of VRPs, does indeed manifest in other combinatorial optimization (CO) problems, such as the Knapsack Problem (KP) with its capacity limit and the Bin Packing Problem (BPP) with bin capacities.
>
> To validate this, we conduct a case study on KP. Following the setup of POMO, we generated KP200 instances with varying capacities: $C \in$ {2, 5, 10, 25, 50}. We then evaluate two representative models, POMO and BQ, using their publicly available checkpoints that are trained specifically on KP200 instances with a fixed capacity of C=25. Both models used greedy search for inference. The results are presented in Table 11.
>
>
> Table 11: Performance of baseline NCO models on KP200 instances with different capacities. The larger the value is, the better the model performs.
> ||KP200||||||||||
> |:-:|:-:|:-:|:-:|:-:|:-:|:-:|:-:|:-:|:-:|:-:|
> ||C=2||C=5||C=10||C=25||C=50||
> |Method|Value|Gap|Value|Gap|Value|Gap|Value|Gap|Value|Gap|
> |OR-Tools|16.157|0.000\%|25.723|0.000\%|36.467|0.000\%|57.665|0.000\%|81.163|0.000\%|
> |POMO|9.673|40.132\%|23.539|8.488\%|34.428|5.590\%|57.393|**0.472\%**|79.855|1.611\%|
> |BQ|15.653|3.119\%|25.546|0.686\%|36.369|0.269\%|57.611|**0.094\%**|80.624|0.664\%|
>
> As shown in the table, the performance of both POMO and BQ degrades significantly as the capacity C deviates from the training value of 25. For the tight constraint C=2, the POMO model's gap explodes to 40.132\%, an 85$\times$ increase compared to its performance at C=25. Similarly, the BQ model's gap at C=2 is 33$\times$ larger than its gap at C=25. When the constraint becomes looser at C=50, the performance drop is also evident, with the gaps for POMO and BQ increasing by 3.4$\times$ and 7$\times$, respectively. These experimental results on KP strongly demonstrate that the constraint overfitting issue also occurs in other CO problems, extending beyond VRPs.
>
>
> Thank you again for your time and effort dedicated to reviewing our work. We will carefully incorporate the above discussions into our revised paper. We sincerely hope that our responses can effectively address your concerns.

---

> > ### Comment · Reviewer_acR7 · 2025-08-02
> >
> > Thank you for your comprehensive rebuttal, which has contributed valuable discussions. I hope the new findings regarding overfitting on the demand distribution and the KP problem will be incorporated into the final version of your paper, as they indeed enhance the broader impact of your work.
> >
> > With my concerns now fully addressed, I am pleased to raise my score to a clear acceptance.

---

> > > ### Author Response · Authors · 2025-08-07
> > > **Thank you very much**
> > >
> > > Dear Reviewer acR7,
> > >
> > > Thank you very much for your effort in reviewing our paper and engaging with us in the discussion. We are glad to know your concerns are fully addressed, and you have increased the score to a clear acceptance. We will carefully incorporate the discussion regarding the overfitting issue on the demand distribution and the KP problem in the revised paper.
> > >
> > > Best Regards,
> > >
> > > Submission 22722 Authors

---

### Official Review · Reviewer_H9mr · 2025-06-28

**Clarity:** 1
**Significance:** 2
**Originality:** 2
**Rating:** 4
**Confidence:** 3

**Summary:**

This study focuses on capacitated vehicle routing problems (CVRP) and aims to improve the generalization performance of neural solvers with respect to the tightness of capacity constraints. Based on the hypothesis that existing methods fail to generalize because they are trained only on instances with a single capacity constraint, the authors propose training on instances with various capacity constraints. Experimental results demonstrate that this approach improves generalization performance regarding constraint tightness. Additionally, the introduction of a multi-expert module is shown to produce solutions with shorter route lengths.

**Questions:**

I would appreciate it if the authors could answer the following questions.
- Could you provide a formal definition of constraint tightness as used in the paper?

- Besides the experimental results presented, do you have other evidence supporting the claim that existing methods overfit to capacity constraints?

- Between VCT and MEM, which component contributes more to the performance improvement?

**Ethical Concerns:**

["NO or VERY MINOR ethics concerns only"]

**Final Justification:**

This paper addresses the issue of overfitting to the degree of constraint tightness present in the training data, an important limitation of neural network–based combinatorial optimization solvers.
The proposed method is simple, yet its effectiveness is demonstrated empirically.
My initial concerns regarding the definition of constraint tightness, the inclusion of ablation studies, and the clarity of the problem setting have been largely resolved through the rebuttal process.

**Limitations:**

yes

**Quality:**

2

**Strengths And Weaknesses:**

**Strengths**

- The paper demonstrates that a simple approach of changing the training data from instances with a single capacity constraint to instances with multiple capacity constraints improves generalization performance with respect to capacity constraint tightness.

- Based on the observation that the structure of solutions differs depending on the tightness of capacity constraints, the authors introduce a multi-expert module and show that it further shortens route lengths beyond what can be achieved by changing the training data alone.

**Weaknesses**

- The paper has several issues regarding clarity. First, definitions of key terminology are missing. For example, “constraint tightness,” a central concept in the paper, is not clearly defined. Second, the problem setting itself is not clearly described. In many CVRP formulations, the number of vehicles is part of the problem instance (i.e., given). In such settings, feasibility cannot be determined solely based on vehicle capacity. This affects the observation in Section 4.1 that solution characteristics change depending on constraint tightness.

- The paper lacks sufficient experimental justification for why existing methods have limited generalization performance with respect to constraint tightness. For example, an analysis of whether the solution structure (e.g., the number of vehicles used, or the number of nodes visited per vehicle) changes significantly when the capacity varies would strengthen the claim that existing models are overfitting to capacity constraints.

- The ablation study is incomplete. Table 4 reports results for the base model, base model + VCT, and base model + VCT + MEM, but does not include results for the base model + MEM. To clarify the contribution of VCT versus MEM, I recommend reporting the performance of the base model + MEM as well.

---

> ### Author Rebuttal · Authors · 2025-07-31
>
> Dear Reviewer H9mr,
>
> Thank you very much for taking the time and effort to review our work. We are delighted to know that you find our paper to demonstrate a simple approach that can improve model generalization performance with respect to constraint tightness, and that our introduction of a multi-expert module further provides additional performance gains.
>
> We address your concerns point-by-point as follows.
>
> > **C1. The paper has several issues regarding clarity. First, definitions of key terminology are missing. For example, “constraint tightness,” a central concept in the paper, is not clearly defined.**
>
> > **Q1. Could you provide a formal definition of constraint tightness as used in the paper?**
>
> Thank you very much for these valuable comments. We apologize for not providing a clear, formal definition of constraint tightness in the submitted version. In response to your comment, we would like to provide the formal definition as follows.
>
> **1. Definition of Constraint Tightness:**
>
> In this paper, “constraint tightness” is a metric used to quantify the degree of restriction imposed by a specific constraint in a combinatorial optimization problem.
>
> A higher degree of tightness implies that the constraint more severely limits the feasible solution space, making the problem more restricted. For example, in CVRP, an extremely tight constraint—such as a capacity so small that a vehicle can only serve one customer—dramatically shrinks the solution space to a single, fixed solution. Conversely, a lower degree of tightness implies that the constraint is less restrictive, allowing for greater flexibility in the solution space. For example, when the vehicle capacity is large enough to serve all customers, this loose constraint effectively transforms the problem into TSP, causing the solution space to explode to a factorial size with respect to the total number of customer nodes.
>
> To be more specific, we implement this concept for the two vehicle routing problems studied in our paper as follows:
>
> **2. For the Capacitated Vehicle Routing Problem (CVRP):**
>
> The “constraint tightness” is directly represented by the vehicle capacity value, $C$.
>
> + A tight constraint corresponds to a low capacity value (e.g., $C=10$ in our paper). In this scenario, each vehicle can only serve a very limited number of customers, forcing the solution to be composed of many short sub-tours.
> + A loose constraint corresponds to a high capacity value (e.g., $C=500$). Here, the capacity is so large that it is barely a limiting factor, allowing a single vehicle to serve many or all customers. This makes the problem characteristics approach those of TSP.
> + Therefore, in the context of CVRP, constraint tightness is inversely proportional to the vehicle capacity $C$.
>
> **3. For the CVRP with Time Windows (CVRPTW):**
>
> The “constraint tightness” refers to the strictness of the customer service time windows. As detailed in Appendix E of our paper, we introduce a coefficient, $\alpha$, to quantify this.
>
> + We use the following equation to relax or tighten each customer's original time window $[e_i, l_i]$ to the new time window $[e_i',l_i']$: $  e_i' = \max(e_i + \delta,0),l_i' = l_i - \delta, \text{where} \ \delta = \frac{l_i-e_i}{2} \cdot (1 - \alpha)$.
> + A tight constraint corresponds to a small $\alpha$ value (e.g., $\alpha=0.2$). This significantly narrows the permissible service time window for each customer, increasing the difficulty of finding a feasible route.
> + A loose constraint corresponds to a large $\alpha$ value (e.g., $\alpha=3.0$). This relaxes the time window constraints, providing more flexibility in routing.
> + Therefore, for CVRPTW, the tightness of the time window constraint is inversely proportional to the coefficient $\alpha$.
>
> > **W2. Second, the problem setting itself is not clearly described. In many CVRP formulations, the number of vehicles is part of the problem instance (i.e., given). In such settings, feasibility cannot be determined solely based on vehicle capacity. This affects the observation in Section 4.1 that solution characteristics change depending on constraint tightness.**
>
> Thank you very much for this valuable comment. We apologize for not making the number of vehicles in the problem setting sufficiently clear. We would like to clarify that our work assumes an unlimited fleet of homogeneous vehicles for each problem instance, which follows the classical VRP formulation used in seminal NCO papers [1, 2]. We will carefully revise Section 2.1 (Problem Definition) in our paper to explicitly state this assumption.
>
> [1] Attention, learn to solve routing problems! ICLR, 2019.
>
> [2] Pomo: Policy optimization with multiple optima for reinforcement learning. NeurIPS, 2020.
>
> > **W3. The paper lacks sufficient experimental justification for why existing methods have limited generalization performance with respect to constraint tightness. For example, an analysis of whether the solution structure (e.g., the number of vehicles used, or the number of nodes visited per vehicle) changes significantly when the capacity varies would strengthen the claim that existing models are overfitting to capacity constraints.**
>
> > **Q2. Besides the experimental results presented, do you have other evidence supporting the claim that existing methods overfit to capacity constraints?**
>
> Thank you very much for your insightful and constructive comment. To provide other evidence supporting our claim, we conduct an analysis focusing on a key aspect of the solution structure: the average number of vehicles used. We compare the solutions generated by three representative neural solvers (POMO, BQ, and LEHD), all pre-trained on CVRP100 instances with C=50, against the (near)-optimal solutions from the classical HGS solver. The evaluation is performed on datasets with varying capacities (C = 10, 50, 100, 500). The results presented in Table 5 show the average number of vehicles used by each method and the percentage gap in the number of vehicles relative to the HGS baseline.
>
> Table 5: Comparison of average number of vehicles used across different capacities
> ||C=10||C=50||C=100||C=500||
> |:-:|:-:|:-:|:-:|:-:|:-:|:-:|:-:|:-:|
> |Method|Avg. Vehicle Num.|Vehicle Num. Gap|Avg. Vehicle Num|Vehicle Num. Gap|Avg. Vehicle Num|Vehicle Num. Gap|Avg. Vehicle Num|Vehicle Num. Gap|
> |HGS|53.19|0.00\%|10.44|0.00\%|5.44|0.00\%|1.44|0.00\%|
> |POMO|57.18|7.50\%|10.66|**2.11\%**|6.21|14.15\%|3.73|159.03\%|
> |BQ|56.84|6.86\%|10.62|**1.72\%**|5.61|3.13\%|1.91|32.64\%|
> |LEHD|65.43|23.01\%|10.95|**4.89\%**|5.86|7.72\%|1.94|34.72\%|
>
> From these results, we can observe that the gap in the average number of vehicles used by the neural models, relative to the HGS baseline, is minimized at C=50. As the capacity deviates from this training value, i.e., becoming either tighter (C=10) or looser (C=100, 500), the gap increases substantially. For example, the gap for LEHD at C=10 and C=500 is 4.7x and 7.1x larger, respectively, than its gap at C=50.
> This demonstrates that when faced with instances requiring a fundamentally different solution structure (e.g., $\approx$53 vehicles for C=10 or $\approx$1 vehicles for C=500), the models fail to adapt. They generate solutions with a suboptimal number of vehicles, which directly contributes to the higher overall travel costs reported in our main paper.
>
>
> > **W3. The ablation study is incomplete. I recommend reporting the performance of the base model + MEM as well.**
>
> > **Q3. Between VCT and MEM, which component contributes more to the performance improvement?**
>
> Thank you very much for this constructive comment. We agree with you that including the "Base Model + MEM" configuration is essential for a complete ablation study and for clearly isolating the individual contributions of VCT and MEM.
>
> Following your suggestion, on top of Table 4 in the main paper, we have added the ablation study on the effect of MEM. We train the base model with the Multi-Expert Module (MEM) but without Varying Constraint Tightness (VCT) training, that is, the model was trained exclusively on CVRP100 instances with a fixed capacity of C=50. We then evaluated its performance on the same test sets. The complete results are presented in the updated table below.
>
> From these comprehensive results, we can observe that:
>
> + **Base Model + MEM:** Adding only the MEM reduces the average gap from 10.78\% to 8.95\%. This shows that the MEM provides a modest benefit, likely by improving performance on the in-domain (C=50) and nearby capacities.
> + **Base Model + VCT:** Adding only VCT training leads to a dramatic improvement, reducing the average gap from 10.78\% to just 2.41\%. This demonstrates that exposing the model to a wide range of constraint tightnesses during training is the primary driver of generalization.
>
>
> Therefore, these results clearly show that VCT contributes significantly more to the overall performance improvement than MEM does. The role of MEM is to further refine the generalized policy learned via VCT, leading to the best overall performance (1.86\% average gap).
>
> Table 6: Effects of Varying Constraint Tightness training (VCT) and Multi-Expert Module (MEM). The Base Model is only trained on instances with C=50 and without MEM.
> ||CVRP100||||||||
> |-|:-:|:-:|:-:|:-:|:-:|:-:|:-:|:-:|
> ||C=10|C=50|C=100|C=200|C=300|C=400|C=500|Varying Capacities|
> ||Gap|Gap|Gap|Gap|Gap|Gap|Gap|Avg.Gap|
> |Base Model|45.64\%|3.72\%|4.21\%|3.62\%|5.61\%|5.36\%|7.29\%|10.78\%|
> |Base Model+MEM|37.03\%|**3.28\%**|4.41\%|2.90\%|4.56\%|4.39\%|6.11\%|8.95\%|
> |Base Model+VCT|1.88\%|4.51\%|4.69\%|2.25\%|1.28\%|1.03\%|1.25\%|2.41\%|
> |Base Model+VCT+MEM|**1.54\%**|3.82\%|**3.67\%**|**1.49\%**|**0.87\%**|**0.75\%**|**0.87\%**|**1.86\%**|
>
>
> Thank you again for your time and effort dedicated to reviewing our work. We will carefully incorporate the above discussions into our revised paper. We sincerely hope that our responses can effectively address your concerns.

---

> > ### Comment · Reviewer_H9mr · 2025-08-05
> >
> > Thank you for providing the additional experimental results and insights. I find that most of my initial concerns have been satisfactorily addressed.
> >
> > However, I still have concerns about the definition of constraint tightness. The provided definition appears to be highly problem-specific and lacks sufficient generality. In the case of CVRP, tightness can be defined on a per-instance basis, enabling meaningful comparisons across problem instances. However, in CVRPTW, tightness is defined in relative terms with respect to other instances, making it difficult to quantify the tightness of a given instance in absolute terms. As a result, comparing tightness across problems with significantly different time window constraints becomes problematic.
> >
> > Given that constraint tightness constitutes a central concept in this work, I believe it merits a more rigorous and general formulation. I would be interested to hear the authors’ thoughts on how tightness might be defined in a more abstract and problem-independent manner.

---

> ### Author Response · Authors · 2025-08-06
> **Response Regarding the Definition of "Constraint Tightness" (1/3)**
>
> Dear Reviewer H9mr,
>
> Thank you very much for your valuable comment. We fully agree that a more rigorous and general definition of "constraint tightness" is important. Your insightful comments have prompted us to refine our approach, leading to a more general definition that is independent of specific problem features as follows.
>
> We propose to define the "**tightness**" of a constraint as: **the degree of degradation in the objective value of a solution produced by a reference algorithm due to the introduction of that constraint.**
>
> The intuition is that tighter constraints restrict the solution space, forcing a reference algorithm to find higher-cost solutions. That is, a constraint acts as an impediment to the algorithm, forcing it to construct a less direct, more complex solution path. This manifests as a quantifiable increase in the final objective value. For example, in CVRP, a tight capacity constraint (e.g., $C=10$) necessitates frequent returns to the depot, resulting in many short, high-cost sub-tours. This contrasts with a loose constraint (e.g., $C=500$), where the solution structure resembles a single, efficient TSP tour.
>
> In this paper, we use simple, general heuristics like Nearest Neighbor (NN) as reference algorithms. We avoid using strong solvers to ensure that our definitions are both practically simple and general. Furthermore, simple heuristics are computationally inexpensive, facilitating an efficient analysis. Moreover, a simple heuristic method is easy to design for most of the CO problems, ensuring the good generalization of our definition.
>
> **1. Formal Definition of Constraint Tightness**
>
> We formalize this definition as follows. Let $P$ be a combinatorial optimization (CO) problem instance, with the objective function $f(\pi)$, where $\pi$ is a solution to the instance:
>
> * Let $P_\xi$ be the instance with a specific constraint $\xi$.
> * Let $P_{\emptyset}$ be the unconstrained baseline, representing the problem with constraint $\xi$ fully relaxed.
> * Let $P_1$ be the extreme-constraint version, where the constraint $\xi$ is exceedingly tight, serving as an upper bound for the solution objective value degradation.
> * Let $\mathcal{H}$ be a simple, general deterministic heuristic solving algorithm. It serves as the reference algorithm.
> * Let $f(\mathcal{H}(P))$ be the objective value of the solution generated by the algorithm $\mathcal{H}$ for an instance $P$.
>
> Then, the tightness $T(\xi)$ of the constraint $\xi$ can be defined as:
>
> $$T(\xi, P, \mathcal{H})=\frac{log(f\left(\mathcal{H}\left(P_\xi\right)\right))-log(f\left(\mathcal{H}\left(P_{\emptyset}\right)\right))}{log(f\left(\mathcal{H}\left(P_1\right)\right))-log(f\left(\mathcal{H}\left(P_{\emptyset}\right)\right))}$$
>
> **Interpretation of the Formula:**
>
> * **This formula calculates a normalized rate of the objective value degradation caused by constraint $\xi$.** The numerator measures the logarithmic increase in the objective value relative to the unconstrained baseline $P_{\emptyset}$. This is then normalized by the denominator, which measures the maximum possible logarithmic increase in the objective value from the "unconstrained" to the "extreme" case.
> * The resulting tightness score, **$T(\xi)$, is a dimensionless value within the range [0,1]**. A score of $T(\xi)=0$ signifies a completely "loose" constraint with no negative impact on the solution. $T(\xi)=1$ indicates that the constraint imposes the maximum possible solution objective value degradation, equivalent to the extreme case.
> * **Logarithmic Scaling:** We observe that objective function values in CO problems can change sharply as the constraint tightness changes (as shown in the table below). Therefore, we employ the logarithm to compress this scale, making the tightness metric more stable.
>
> **This formula offers several key advantages:**
>
> 1.  **Generality:** It quantifies tightness by observing the impact of the constraint on the solution generated by a general heuristic algorithm, rather than relying on problem-specific physical parameters (such as capacity, time window width, etc.). It is therefore applicable to any combinatorial optimization problem for which "unconstrained" and "extreme" baseline cases can be defined.
> 2. **Absolute Quantifiability:** It provides a dimensionless scalar value for any given instance, enabling meaningful tightness comparisons across different instances.
> 3. **Reproducibility**: By using the simple and general heuristic as a reference algorithm, the tightness calculation is fully reproducible and computationally inexpensive.
>
> While developing this definition, we also considered alternative definitions, such as those based on solution structure (e.g., number of sub-tours) or constraint features (e.g., capacity values). These were found to be either too problem-specific or difficult to generalize. We therefore selected defining tightness via its impact on the objective function value, as mentioned above.

---

> ### Author Response · Authors · 2025-08-06
> **Response Regarding the Definition of "Constraint Tightness" (2/3)**
>
> **2. Generalization to Different Problems**
>
> To demonstrate the versatility of this general definition, we apply it to three CO problems: two VRP variants (CVRP and CVRPTW) and a non-VRP problem - the Knapsack Problem (KP) as follows.
>
> **2.1 For CVRP:**
>
> * Constrained Instance ($P_\xi$): A standard CVRP instance with a finite vehicle capacity $C_{cap}$.
> * Unconstrained Baseline ($P_{\emptyset}$): This baseline is formed by relaxing the capacity value to a sufficiently large value to serve all customers in a single tour.
> * Extreme Constraint Baseline ($P_1$): This baseline represents the most restrictive scenario, where constraints are so tight (e.g., capacity is minimal) that each vehicle can serve only one customer before returning to the depot.
> * Reference Algorithm ($\mathcal{H}$): We use a Nearest Neighbor (NN) heuristic adapted for the CVRP. The algorithm iteratively constructs routes by adding the closest unvisited customer. If adding a customer would violate the vehicle's remaining capacity, the current route is finalized, and a new route begins from the depot.
>
> **2.2 For CVRPTW:**
>
> * Constrained Instance ($P_\xi$): A standard CVRPTW instance with time window constraints.
>
> * Unconstrained Baseline ($P_{\emptyset}$): This baseline is derived by relaxing the time windows to be infinitely large.
>
> * Extreme Constraint Baseline ($P_1$): Similar to the CVRP case, extremely tight time windows can force a vehicle to return to the depot after each delivery, resulting in single-customer routes.
>
> * Reference Algorithm ($\mathcal{H}$):  The reference heuristic is an adapted Nearest Neighbor (NN) algorithm. It extends the CVRP's NN logic by verifying both capacity and time window feasibility before appending a customer to a route.
>
> **2.3 For the Knapsack Problem (KP):**
>
> * Problem Description: Given a set of items, each with a weight and a value randomly sampled from (0, 1), select which items to include in a knapsack so that the total weight is less than or equal to a given limit weight capacity $C_{kp}$. The objective is to maximize the total value of items in the knapsack. The instances used in this study, denoted as KP200, consist of 200 items.
>
> * Constrained Instance ($P_\xi$): A standard KP instance with a knapsack capacity constraint $C_{kp}$.
>
> * Unconstrained Baseline ($P_{\emptyset}$): This corresponds to a knapsack with infinite capacity. The optimal solution is trivial: select all items.
>
> * Extreme Constraint Baseline ($P_1$): This occurs when the knapsack capacity is set to the minimum weight among all available items. Consequently, the only feasible non-empty solutions involve selecting the single, lightest item.
>
> * Reference Algorithm ($\mathcal{H}$): The reference is a standard greedy algorithm. It sorts items by their value-to-weight ratio in descending order and sequentially adds them to the knapsack until no more items can fit within

---

> ### Author Response · Authors · 2025-08-06
> **Response Regarding the Definition of "Constraint Tightness" (3/3)**
>
> **3. Experimental Validation**
>
> We evaluated the proposed tightness formula across the CVRP, CVRPTW, and KP instances.
> The results, summarized in Table A1 below, reveal a clear and consistent trend: as constraints are relaxed from the extreme case towards the unconstrained baseline, **the calculated Tightness value smoothly transitions from 1 to 0**. For example, in the CVRP instances, increasing the vehicle capacity ($C_{cap}$) from 10 to 50 and then to 500 causes the tightness to decrease from 0.837 to 0.280 and 0.018, respectively, bridging the gap between the extreme case (Tightness value = 1.0) and the unconstrained TSP (Tightness value = 0.0).
>
> These findings confirm that our metric provides a general and robust measure of constraint tightness. It successfully quantifies the absolute tightness of a single instance while also enabling direct comparison of relative tightness between different instances.
>
> In addition, we would like to clarify that the analysis in our main paper explores a comprehensive range of tightness values, i.e., [0.018, 0.837] for CVRP with $C\in[10,500]$ and [0.072, 0.717] for CVRPTW with $\alpha \in [0.2, 3]$. This scope is broad enough to reflect the overfitting issue associated with the NCO model trained on CVRP100 (tightness = 0.280) and CVRPTW (tightness = 0.418).
>
> We sincerely appreciate your time and effort in this discussion. We hope our response can address your concerns. Please do not hesitate to let us know if you have any further questions, and we will do our best to provide a prompt response.
>
> Table A1: Tightness Value Calculation Results for Different Problems under Various Degrees of Constraint Tightness. "Extreme Case" means that the constraint is exceedingly tight, serving as an upper bound for the solution objective value degradation.
> |||CVRP100|||||||||
> |:-:|:-:|:-:|:-:|:-:|:-:|:-:|:-:|:-:|:-:|:-:|
> |Method||Extreme Case|C=10|C=50|C=100|C=200|C=300|C=400|C=500|Unlimited $C_{cap}$|
> |Nearest Neighbor|Obj. Value $\downarrow$|103.986|70.684|18.886|13.774|11.329|10.415|10.494|10.146|9.726|
> ||Tightness Value|1.000|0.837|0.280|0.147|0.064|0.029|0.032|0.018|0.000|
> |||CVRPTW100|||||||||
> |||Extreme Case|$\alpha$=0.2|$\alpha$=0.5|$\alpha$=1|$\alpha$=1.5|$\alpha$=2|$\alpha$=2.5|$\alpha$=3|Unlimited $\alpha$|
> |Nearest Neighbor|Obj. Value $\downarrow$|103.986|64.422|51.709|40.473|30.436|25.687|23.043|21.611|19.136|
> ||Tightness Value|1.000|0.717|0.587|0.443|0.274|0.174|0.110|0.072|0.000|
> |||KP200|||||||||
> |||Extreme Case|C=0.1|C=1|C=2|C=5|C=10|C=25|C=50|Unlimited $C_{kp}$|
> |Greedy|Obj. Value $\uparrow$|0.532|3.346|11.217|16.103|25.743|36.477|57.655|81.116|99.865|
> ||Tightness Value|1.000|0.649|0.418|0.349|0.259|0.192|0.105|0.040|0.000|

---

> > ### Comment · Reviewer_H9mr · 2025-08-07
> >
> > Thank you very much for your detailed response.
> > The definition of tightness you proposed is highly general, and I agree with the advantages you highlighted in your reply.
> > Furthermore, the experimental results you presented convincingly demonstrate that the definition aligns well with the intuitive notion of constraint tightness.
> > With my major concerns now largely resolved, I am pleased to raise my score to 4.
> >
> > As a minor suggestion, I would appreciate it if the following points could be made explicit in the camera-ready version:
> >
> > * The range of the function $f$ is strictly greater than 0.
> > * The heuristic H can find a feasible solution to the problem $P$.
> > * The value of $f$ used to compute tightness refers to the objective value of a feasible solution.

---

> > > ### Author Response · Authors · 2025-08-07
> > > **Thank you very much**
> > >
> > > Dear Reviewer H9mr,
> > >
> > > Thank you very much for your effort in reviewing our paper and engaging with us in the discussion. We are glad to know your major concerns are largely resolved, and you have increased the score to 4. We will carefully and explicitly incorporate the points you mentioned in the revised paper.
> > >
> > > Best Regards,
> > >
> > > Submission 22722 Authors

---

### Official Review · Reviewer_tyK7 · 2025-07-02

**Clarity:** 3
**Significance:** 2
**Originality:** 3
**Rating:** 3
**Confidence:** 4

**Summary:**

This paper investigates a critical, yet often overlooked, challenge in Neural Combinatorial Optimization (NCO) for Vehicle Routing Problems (VRPs): the impact of constraint tightness on model performance. Most existing NCO methods train and test on data with fixed constraint values, leading to a strong tendency to overfit specific constraint degrees. This results in significant performance degradation when applied to problems with different (i.e., looser or tighter) constraints, limiting their real-world applicability.

Taking the Capacitated Vehicle Routing Problem (CVRP) as a primary example, the paper empirically demonstrates this overfitting issue. To address this drawback, the authors propose a novel training scheme and a multi-expert module designed to learn a more generally adaptable solving strategy.

The key contributions of this work are:

Empirical Analysis of Constraint Overfitting: The paper provides a thorough empirical analysis demonstrating that existing NCO methods for CVRP overfit the capacity constraint, performing well only within a narrow range of constraint values.

Efficient Training Scheme for Varying Tightness: A new training scheme is introduced that explicitly considers and incorporates varying degrees of constraint tightness during the training phase.

Multi-Expert Module: A novel multi-expert module is proposed. This module learns to adapt its solving strategy based on the specific constraint tightness of a given problem instance, making the overall model more robust and generalizable.

Improved Generalization: Extensive experimental results on CVRP and CVRP with Time Windows (CVRPTW) with diverse constraint tightness degrees show that the proposed method effectively overcomes the overfitting issue and achieves superior performance across a wide range of constraint values.

**Questions:**

Question: The paper proposes a "multi-expert module" to learn a generally adaptable solving strategy. However, could the authors elaborate on whether the model truly learns a continuous, unified generalization across the entire spectrum of constraint tightness, or if its adaptation primarily involves switching between implicitly specialized "experts" optimized for specific ranges of tightness?

Question: The definition of "constraint tightness" is effectively applied to CVRP and CVRPTW. However, how can this specific definition and the proposed multi-expert framework be universally applied or adapted to other types of Combinatorial Optimization Problems (COPs) or more diverse VRP variants with qualitatively different or more complex constraint structures (e.g., time windows for specific deliveries, vehicle type constraints, dynamic capacities)?

**Ethical Concerns:**

["NO or VERY MINOR ethics concerns only"]

**Limitations:**

Yes.

**Paper Formatting Concerns:**

No.

**Quality:**

3

**Strengths And Weaknesses:**

Strengths:

Quality: The paper addresses a highly relevant and often overlooked problem in NCO: generalization across different constraint tightness degrees. The empirical analysis demonstrating overfitting in existing methods is well-conducted and clearly illustrates the problem. The proposed solution, involving both a novel training scheme and a multi-expert module, is technically sound and logically designed to tackle the identified issue. The experimental validation is comprehensive, testing on CVRP and CVRPTW with varying capacities and time window tightness, showing consistent and significant improvements over strong baselines.

Clarity: The paper is well-written and structured. The problem of constraint overfitting is clearly defined and motivated. The empirical analysis supporting this problem is presented concisely with illustrative figures. The proposed training scheme and the multi-expert module are explained logically, with clear diagrams (e.g., Figure 2) that aid in understanding the architecture and flow. The mathematical formulations are clear, and the overall narrative flows smoothly, making it easy to comprehend complex ideas.

Significance: This paper makes a significant contribution by highlighting and providing an effective solution for the constraint overfitting problem in NCO. This issue is a major barrier to the practical deployment of NCO models, as real-world problems rarely conform to a single, fixed constraint value. By enabling NCO models to generalize across different constraint tightness degrees, the work significantly enhances the robustness, flexibility, and real-world applicability of neural solvers for VRPs. The insight into "rethinking" how NCO models are trained with respect to constraints is impactful.

Originality: The core idea of systematically analyzing and then addressing the problem of NCO overfitting to specific constraint tightness degrees is highly original. While multi-task learning or conditional models exist, the specific formulation of the "constraint tightness embedding" and the design of the "multi-expert module" to adapt the solving strategy based on this embedding are novel contributions within the NCO domain. The empirical evidence of this overfitting phenomenon is also an original insight that challenges common NCO training paradigms.

Weaknesses:
While the paper proposes a "multi-expert module" to learn a generally adaptable solving strategy, a weakness is that this module, in practice, function more akin to an ensemble of specialized sub-models, each implicitly optimized for a specific range of constraint tightness. If the adaptation primarily involves switching between these specialized "experts" rather than a continuous, unified generalization across the entire spectrum of tightness, it could diminish the originality of achieving a single, truly adaptable policy and might be seen as a sophisticated form of training multiple models for different tightness regimes.

The specific definition of "constraint tightness", while effective for the Capacitated Vehicle Routing Problem (CVRP) and CVRP with Time Windows (CVRPTW), might be too problem-specific. This could limit the universality and direct applicability of the proposed multi-expert architecture and training scheme to other types of Combinatorial Optimization Problems (COPs) or even more diverse VRP variants with different or more complex constraint structures, thus potentially reducing its overall generalizability beyond the explored VRP contexts.

---

> ### Author Rebuttal · Authors · 2025-07-31
>
> Dear Reviewer tyK7,
>
> Thank you very much for taking the time and effort to review our work.
> We are delighted to know you find our paper well-written and structured, makes a significant contribution by highlighting and providing an effective solution for the constraint overfitting problem in NCO, and our core idea is highly original.
>
> We address your concerns point-by-point as follows.
>
>
> > **C1. While the paper proposes a "multi-expert module" to learn a generally adaptable solving strategy, a weakness is that this module, in practice, functions more akin to an ensemble of specialized sub-models, each implicitly optimized for a specific range of constraint tightness. If the adaptation primarily involves switching between these specialized "experts" rather than a continuous, unified generalization across the entire spectrum of tightness, it could diminish the originality of achieving a single, truly adaptable policy and might be seen as a sophisticated form of training multiple models for different tightness regimes.**
>
> > **The paper proposes a "multi-expert module" to learn a generally adaptable solving strategy. However, could the authors elaborate on whether the model truly learns a continuous, unified generalization across the entire spectrum of constraint tightness, or if its adaptation primarily involves switching between implicitly specialized "experts" optimized for specific ranges of tightness?**
>
>
> Thank you very much for your insightful comment and comment and question. We acknowledge that our multi-expert module, in its current form, functions by switching between specialized experts rather than learning a single, continuously adaptable policy. This is a deliberate design choice informed by empirical investigation, and we appreciate the opportunity to elaborate on our reasoning.
>
> The goal of learning a unified policy that generalizes continuously across the entire constraint spectrum is indeed highly desirable. We explore this direction by implementing a top-K Mixture-of-Experts (MoE) model with a learnable and soft gating network following MVMoE [1], with $K=2$. This approach replaces the rule-based switching with a mechanism that learns to assign weights to different experts, allowing for a weighted combination of their outputs based on the input features. Due to space limitations, we will detail the implementation of the top-K MoE module in the revised paper.
>
> We train the model with this Top-K MoE module using the same settings as our proposed method (detailed in Section 5) and compare it against the switching mechanism ("Switching"). The models are evaluated on CVRP100 instances with the capacity values $C \in${10,50,100,200,300,400,500}. As shown in Table 3, while the Top-K MoE approach is conceptually appealing, the simpler switching mechanism achieved superior performance across most tightness regimes, resulting in a lower overall average gap (Switching: 1.86\% vs. Top-K: 2.17\%). Given these empirical results, we select the more performant switching mechanism.
>
>
> Table 3: Comparison of our switching mechanism against a learnable Top-K MoE gating mechanism on CVRP100 instances
> ||CVRP100||||||||
> |:---:|:---:|:---:|:---:|:---:|:---:|:---:|:---:|:---:|
> ||C=10|C=50|C=100|C=200|C=300|C=400|C=500|Varying Capacities|
> ||Gap|Gap|Gap|Gap|Gap|Gap|Gap|Avg.Gap|
> |Top-K|**1.48\%**|4.58\%|4.49\%|1.94\%|1.07\%|0.77\%|**0.83\%**|2.17\%|
> |Switching|1.54\%|**3.82\%**|**3.67\%**|**1.49\%**|**0.87\%**|**0.75\%**|0.87\%|**1.86\%**|
>
>
> Furthermore, we would like to respectfully emphasize that a primary contribution of this work is the identification and systematic analysis of the constraint tightness-overfitting problem in existing NCO methods. While our proposed solution is straightforward, it effectively addresses this newly highlighted issue. We believe our finding and method provide valuable insights that can inspire interesting future works, including the development of more sophisticated, continuously generalized policies.
>
> [1] MVMoE: Multi-Task Vehicle Routing Solver with Mixture-of-Experts, ICML2024.
>
>
>
>
>
>
> > **C2. The specific definition of "constraint tightness", while effective for the Capacitated Vehicle Routing Problem (CVRP) and CVRP with Time Windows (CVRPTW), might be too problem-specific. This could limit the universality and direct applicability of the proposed multi-expert architecture and training scheme to other types of Combinatorial Optimization Problems (COPs) or even more diverse VRP variants with different or more complex constraint structures, thus potentially reducing its overall generalizability beyond the explored VRP contexts.**
>
>
> > **Q2.The definition of "constraint tightness" is effectively applied to CVRP and CVRPTW. However, how can this specific definition and the proposed multi-expert framework be universally applied or adapted to other types of Combinatorial Optimization Problems (COPs) or more diverse VRP variants with qualitatively different or more complex constraint structures (e.g., time windows for specific deliveries, vehicle type constraints, dynamic capacities)?**
>
> Thank you very much for your constructive comment. We acknowledge that our current definitions of "constraint tightness" for capacity and time windows are tailored to the specific VRP variants studied in the paper. Nonetheless, we would like to make the following clarification to discuss the generalizability of applying these two constraints and their tightness to other VRP variants.
>
>
> **1. Generalizability of Capacity and Time Windows Constraint Tightness**
>
> We respectfully argue that capacity and time windows are two of the most fundamental and ubiquitous constraints in the entire field of vehicle routing. Their presence is standard in a vast array of VRP variants. Therefore, a method that robustly handles the tightness spectrum of these core constraints already possesses significant potential for broad applicability.
>
>
>
> **2. Empirical Validation on More Complex VRP Variants**
>
>
> To empirically test the universality of our framework, we conducted new experiments on more complex VRP variants: the Vehicle Routing Problem with Backhauls and Time Windows (VRPBTW) and the Vehicle Routing Problem with Backhauls, Duration Limit (L), and Time Windows (VRPBLTW). These problems introduce additional structural constraints (backhaul and duration limit) on top of the standard capacity and time window constraints. We applied our proposed method (VCT training and the MEM module) to the time window constraint in these new, more complex settings, using POMO as the base model. We follow [1] to configure the settings of additional constraints, backhaul, and duration limit.
>
> The results presented in Table 4 demonstrate the effectiveness of our approach. From the table, we can observe that the standard POMO model once again suffers from severe performance degradation as the time window constraint moves to be either tighter or looser than the standard $\alpha=1.0$. In contrast, our enhanced model (POMO + our method) significantly mitigates this issue, maintaining much more stable and superior performance across the entire tightness spectrum.
>
>
>
> Table 4: Performance on Complex VRP Variants (i.e., VRPBTW and VRPBLTW) with Varying Time Window Tightness. "Gap" represents the performance gap between the model and OR-Tools.
> ||VRPBTW100||||||||
> |:---:|:---:|:---:|:---:|:---:|:---:|:---:|:---:|:---:|
> ||alpha=0.2|alpha=0.5|alpha=1|alpha=1.5|alpha=2|alpha=2.5|alpha=3|VaryingTimeWindows|
> ||Gap|Gap|Gap|Gap|Gap|Gap|Gap|Avg.Gap|
> |POMO|18.81\%|13.35\%|**8.46\%**|12.12\%|19.42\%|30.27\%|43.96\%|20.91\%|
> |POMO+Our Method|**13.23\%**|**11.97\%**|9.34\%|**8.67\%**|**9.07\%**|**8.80\%**|**9.11\%**|**10.03\%**|
> ||||||||||
> ||**VRPBLTW100**||||||||
> ||alpha=0.2|alpha=0.5|alpha=1|alpha=1.5|alpha=2|alpha=2.5|alpha=3|VaryingTimeWindows|
> ||Gap|Gap|Gap|Gap|Gap|Gap|Gap|Avg.Gap|
> |POMO|18.90\%|13.32\%|**8.29\%**|11.42\%|17.06\%|23.73\%|31.16\%|17.70\%|
> |POMO+Our Method|**13.00\%**|**11.80\%**|9.09\%|**8.42\%**|**8.92\%**|**8.45\%**|**8.56\%**|**9.75\%**|
>
>
>
> These results suggest that our framework is not a fragile solution limited to simple problems. Instead, it can function as a robust and adaptable sub-system. It can be integrated into solvers for more complex problems to specifically handle the challenge of varying constraint tightness, without conflicting with the logic required for other constraints. We hope this work brings attention to the critical issue of constraint tightness in the NCO community and inspires future research to develop diverse strategies for this challenge across a wider range of combinatorial optimization problems.
>
>
> Thank you again for your time and effort dedicated to reviewing our work. We will carefully add the above important discussion and the new experimental results to our revised paper. We sincerely hope that our responses have effectively addressed your concerns.

---

> > ### Comment · Reviewer_tyK7 · 2025-08-05
> >
> > Thank you for the detailed response. I appreciate your effort in addressing my concerns and for conducting new experiments to provide a more thorough explanation. The additional data and clarification are very helpful. However, in its current form, the solution's reliance on a switching mechanism between specialized experts, while empirically effective, still feels like a partial answer to the problem of a "truly adaptable policy." While I agree that this is a valuable first step, a more fully generalized approach would be a more complete solution. For this reason, my current evaluation stands.

---

> ### Author Response · Authors · 2025-08-06
>
> Dear Reviewer tyK7,
>
> Thank you very much for your valuable feedback. We agree with your perspective that our proposed switching mechanism between specialized experts is an important first step toward a "truly adaptable policy," rather than an ultimate solution. We would like to take this opportunity to further clarify the core contribution and positioning of our work to address your concern.
>
> **Core Contribution: Rethinking and Revealing the Overfitting Issue**
>
> As we emphasized in our introduction and conclusion, the primary goal of this study is not to propose an algorithmically perfect solution to completely overcome the overfitting problem we identified. Instead, the core contribution of our work lies in systematically "rethinking and revealing" the prevalent overfitting phenomenon concerning constraint tightness in the current Neural Combinatorial Optimization (NCO) landscape. We argue that the community has largely overlooked the fact that models trained with a specific constraint value suffer a sharp decline in generalization performance when faced with instances of varying constraint tightness. Through extensive empirical analysis (as illustrated in Section 3), our work clearly exposes the severity of this issue. We believe that identifying and substantiating this problem is, in itself, of crucial value for encouraging the community to establish more comprehensive and robust evaluation standards.
>
> **Our Method: Demonstrate the Possibility to Tackle this Overfitting Issue**
>
> Building on this, our proposed methods, including the varying constraint tightness training scheme and the multi-expert module, are primarily intended as a constructive "proof of concept." We aim to demonstrate that the "constraint overfitting" problem we revealed, while significant, is not insurmountable. Our solution is designed to prove the possibility of developing more adaptive policies and to offer a preliminary yet effective direction for future research, rather than to claim it as the definitive answer. Therefore, we position this paper as an analysis-focused or "rethinking" paper, whose value lies more in diagnosing a critical issue and validating a potential path toward a solution, rather than in providing a perfectly polished algorithm.
>
> We sincerely appreciate your time and effort in this discussion. We hope our response can address your concerns. Please do not hesitate to let us know if you have any further questions, and we will do our best to provide a prompt response.

---

### Official Review · Reviewer_tHuz · 2025-07-05

**Clarity:** 4
**Significance:** 3
**Originality:** 2
**Rating:** 5
**Confidence:** 4

**Summary:**

The paper studies the effect of constraint tightness on neural solvers for vehicle routing problems. It shows that existing approaches overfit to the capacity tightness of the training data and provide low quality solutions if the capacity in test differs from the one used in training. It then studies two approaches for addressing this shortcoming based on data augmentation and using capacity as a feature. The advantage of the proposed approach is shown in numerical experiments.

**Questions:**

I find the experimental results convincing. However, I would like to see the trade-off between computational budget and solution quality more clearly presented. Specifically:
- can you show explicitly the number of training instances used in the ablation study?
- can you show the training time as a function of the constraint tightness?
- can you show the inference time between the different method as well as compared to HGS?

**Ethical Concerns:**

["NO or VERY MINOR ethics concerns only"]

**Limitations:**

See questions.

**Paper Formatting Concerns:**

No major concern.

**Quality:**

3

**Strengths And Weaknesses:**

The paper is well-written. It presents convincingly the limitations of existing NCOs in extensive experiments. I find the analysis of CVRP solutions as a continuum between OVRP and TSP solutions very intuitive, and clearly showing the effect of constraint tightness on solutions. The methodological contributions to address the shortcomings of existing approach are: (1) augmenting the data with varying constraint tightness, and (2) integrating constraint tightness as a feature when computing node probabilities. Interestingly, the first approach is already sufficient to provide significant improvements compare to the baselines.

---

> ### Author Rebuttal · Authors · 2025-07-31
>
> Dear Reviewer tHuz,
>
> Thank you very much for taking the time and effort to review our work. We are delighted to know you find our paper well-written, convincingly presents in solving the limitations of existing NCOs in extensive experiments, the analysis of CVRP solutions as a continuum between OVRP and TSP solutions is very intuitive, and clearly shows the effect of constraint tightness on solutions, and the proposed approach provides significant improvements compared to the baselines.
>
> We address your concerns point-by-point as follows.
>
> > **Q1. can you show explicitly the number of training instances used in the ablation study?**
>
> Thank you very much for raising this question. To ensure a fair and direct comparison in our ablation study (Section 5.2, Table 4 in our main paper), we maintain a consistent training data budget across all experimental settings. Specifically, **each model is trained on a dataset of 1000,000 CVRP100 instances** along with their corresponding labeled solutions. This number is consistent with the main experimental setup detailed in Section 5.1 (line 223).
>
>
> > **Q2. can you show the training time as a function of the constraint tightness?**
>
>
> Thank you very much for raising this question. We would like to clarify that in our framework, **the training time remains almost constant regardless of the constraint tightness**. To empirically validate this, we conduct an experiment. Specifically, we train seven separate LEHD models (with MEM module but without varying constraint tightness training), each on a dataset of one million CVRP100 instances with a distinct, fixed capacity constraint: $C \in ${10, 50, 100, 200, 300, 400, 500}. All other training parameters and the model architecture were held constant across these experiments. The training time per epoch for each model is reported in Table 1.
>
>
> As the results show, the training time is almost constant across different capacity values. This is an inherent characteristic of the constructive NCO models we investigate. The model builds a solution by sequentially selecting one node at each step for a fixed number of steps (i.e., 100 steps for CVRP100). The capacity constraint $C$ is used to check the feasibility of a partial tour and guide the policy network's output, but it does not alter the number of operations required in each forward and backward pass. Therefore, since the number of nodes and the size of the training dataset are fixed, the training time per epoch is unaffected by the tightness of the capacity constraint.
>
>
>
> Table 1: Model Training time associated with different degrees of constraint tightness
>
> |  | C=10 | C=50 | C=100 | C=200 | C=300 | C=400 | C=500 |
> |---|---|---|---|---|---|---|---|
> | Training time per epoch | 2.40h | 2.39h | 2.37h | 2.44h | 2.44h | 2.45h | 2.45h |
>
>
> > **Q3. can you show the inference time between the different method as well as compared to HGS?**
>
>
> Thank you very much for raising this question. We have recorded the inference times for all baseline models as well as HGS, and present them alongside their average optimality gaps in Table 2. For the NCO models, the reported times represent the total duration required to solve a standard test set of 10,000 CVRP100 instances on a single NVIDIA RTX 3090 GPU. It is important to note that HGS runs on a single CPU, and its reported inference time is not directly comparable to the GPU-accelerated times of the learning-based methods.
>
>
> Table 2: Total Inference time for solving 10,000 CVRP100 instances.
> |  | CVRP100 |  |
> |---|:---:|:---:|
> |  | Avg. Gap | Avg. Time |
> | HGS | 0.00 \% | 4.5h |
> | AM | 23.74\% | 2.0s |
> | POMO | 21.72\% | 3.0s |
> | MDAM | 16.75\% | 26.8s |
> | BQ | 7.22\% | 1.8m |
> | LEHD | 9.62\% | 27.0s |
> | ELG | 16.29\% | 9.5s |
> | INViT | 13.28\% | 2.6m |
> | POMO-MTL | 11.41\% | 3.0s |
> | MVMoE | 16.00\% | 12.6s |
> | Ours | **1.86\%** | 57.6s |
>
> The results highlight that our proposed method achieves an effective balance between solution quality and computational efficiency. Compared with the baseline model LEHD, while our method's inference time increases moderately, it yields a substantial 5.2x reduction in the average optimality gap (from 9.62\% to 1.86\%). Furthermore, when compared to other recent methods like BQ and INViT, our model is not only more accurate but also faster. For example, our method reduces the optimality gap by over 3.8x compared to BQ (1.86\% vs. 7.22\%) while being nearly twice as fast (57.6s vs. 109.8s).
>
> Thank you again for your time and effort dedicated to reviewing our work.  We will carefully incorporate the above discussions into our revised paper. We sincerely hope that our responses can effectively address your concerns.

---

### Decision · Program_Chairs · 2025-09-17

**Decision:**

Accept (poster)

**Comment:**

This paper clearly surfaces an overlooked issue in NCO—dependence on constraint tightness—and substantiates it with systematic analyses on CVRP/CVRPTW. The proposed training scheme and multi-expert module are empirically persuasive across variants. Most reviewers are positive (two Accepts, one Borderline-Accept). One reviewer remains cautious about “truly adaptable policy,” but the authors provided additional experiments and clarified the work’s “rethinking” positioning.

Overall, the novelty, empirical rigor, and expected impact meet the NeurIPS bar. Recommend Accept.